# On the effect of upwind emission controls on ozone in Sequoia National Park

Claire E. Buysse[1], Jessica A. Munyan[2], Clara A. Bailey[2], Alexander Kotsakis[3], Jessica A. Sagona[4], Annie Esperanza[5], Sally E. Pusede[2]

[1]Department of Atmospheric Sciences, University of Washington, Seattle, Washington, 98195, USA
[2]Department of Environmental Sciences, University of Virginia, Charlottesville, Virginia, 22904, USA
[3]Department of Earth and Atmospheric Sciences, University of Houston, Houston, Texas, 77204, USA
[4]New Hampshire Environmental Department of Health and Human Services, Division of Public Health Services, Concord, New Hampshire, 03301, USA
[5]National Park Service, Sequoia and Kings Canyon National Parks, Three Rivers, California, 49093, USA

*Correspondence to*: Sally E. Pusede (sepusede@virginia.edu)

**Abstract.** Ozone ($O_3$) air pollution in Sequoia National Park (SNP) is among the worst of any national park in the U.S. SNP is located on the western slope of the Sierra Nevada Mountains, downwind of the San Joaquin Valley (SJV), which is home to numerous cities ranked in the top ten most $O_3$-polluted in the U.S. Here, we investigate the influence of emission controls in the SJV on $O_3$ concentrations in SNP over a 12-yr time period (2001–2012). We show that the export of nitrogen oxides ($NO_x$) from the SJV has played a larger role in driving high $O_3$ in SNP than transport of $O_3$. As a result, $O_3$ in SNP has been more responsive to $NO_x$ emission reductions than in the upwind SJV city of Visalia, and, in SNP, $O_3$ concentrations have declined faster at a higher elevation monitoring station than at a low elevation site nearer to the SJV. We report $O_3$ trends by various concentration metrics but do so separately for when environmental conditions are conducive to plant $O_3$ uptake and for when high $O_3$ is most common, which are time periods that occur at different times of day and year. We find that precursor emission controls have been less effective at reducing $O_3$ concentrations in SNP in springtime, which is when plant $O_3$ uptake in Sierra Nevada forests has been previously measured to be greatest. We discuss the implications of regulatory focus on high $O_3$ days in SJV cities on $O_3$ concentration trends and ecosystem impacts in SNP.

## 1 Introduction

Sequoia National Park (SNP) is a unique and treasured ecosystem that is also one of the most ozone-polluted national parks in the U.S. (National Park Service, 2015a). Ozone ($O_3$) concentrations in SNP exceeded the current U.S. human health-based $O_3$ National Ambient Air Quality Standard (NAAQS), defined as maximum daily 8-h average (MD8A) $O_3$ greater than 70 ppb, on an average of 117 days per year over the time period 2001–2012. While $O_3$ is harmful to humans, it is also damaging to plants and ecosystems (e.g., Reich, 1987), with visible $O_3$ injury observed in many forests across the U.S. (Costonis, 1970; Pronos and Vogler, 1981; Ashmore, 2005), including in SNP (Peterson et al., 1987; Peterson et al., 1991; Patterson and Rundel,

1995; Grulke et al., 1996; National Park Service, 2013). $O_3$ exposure affects ecosystems in a variety of ways, potentially decreasing plant growth (Wittig et al., 2009), reducing photosynthesis and disrupting carbon assimilation (Wittig et al., 2007; Fares et al., 2013), diminishing ecosystem gross and net primary productivity (Ainsworth et al., 2012; Wittig et al., 2009), modifying plant resource allocation (Ashmore, 2005), and impairing stomatal response (Paoletti and Grulke, 2010; Hoshika et al., 2014).

     SNP is home to more than 1,550 plant taxa with numerous plant species found nowhere else on Earth (Schwartz et al., 2013). One endemic species is the giant sequoia (*Sequoiadendron giganteum*), the largest living tree in the world. Large-tree ecosystems like SNP have been shown to be more sensitive to perturbation (Lutz et al., 2012) because ecological functions are provided primarily by a few large trees, rather than many smaller species. Large-diameter trees disproportionately influence patterns of tree regeneration and forest succession (Keeton and Franklin, 2005), carbon and nutrient storage, forest structure and fuel deposition at death, arboreal wildlife habitats and epiphyte communities (Lutz et al., 2012), and water storage (Sillett and Pelt, 2007), which is of critical importance in drought-prone SNP. While mature sequoias are relatively resistant to $O_3$, seedlings are sensitive, with high $O_3$ demonstrated to cause both visible injury and altered plant-atmosphere light and gas exchange (Miller et al., 1994). Giant sequoias grow in mixed-conifer groves with companion species ponderosa pine (*Pinus ponderosa*) and Jeffrey pine (*Pinus jeffreyi*). $O_3$ impacts on these pines have been documented for decades in SNP (Duriscoe, 1987; Pronos and Vogler, 1981) and include early needle loss, reduced growth, decreased photosynthesis, and lowered annual ring width (Peterson et al., 1987; Peterson et al., 1991).

     SNP is located in Central California on the western slope of the Sierra Nevada Mountains downwind of the $O_3$-polluted San Joaquin Valley (SJV) (Figure 1). Previous model estimates of a pollution episode in August 1990 suggest at least half of peak daytime $O_3$ in SNP is produced upwind from anthropogenic precursors (Jacobson, 2001). For the past two decades, regulations have reduced $O_3$ concentrations in the SJV (Pusede and Cohen, 2012). For example, in Fresno, high $O_3$ days, defined as days when the MD8A exceeded 70 ppb, were 50% less frequent in 2007–2010 than ten years earlier (on high temperature days). At the same time, in Bakersfield, high $O_3$ days were 15–40% less frequent (on high temperature days). $NO_x$ emission controls contributed to these decreases (Pusede and Cohen, 2012), with summertime (April–October) daytime (10 am–3 pm local time, LT) nitrogen dioxide ($NO_2$) concentrations falling by 50% from 2001 to 2012, changing linearly by –0.5 ppb yr$^{-1}$ in the SJV city of Visalia. The precursor reductions that brought about these decreases in high $O_3$ are likely to have also affected $O_3$ in SNP.

     The success of $O_3$ regulatory strategies can be measured through attainment of human health-based NAAQS and ecosystem-impact metrics. While there is a secondary NAAQS requirement aimed at vegetation protection, this has historically been the same metric (based on the MD8A $O_3$) at the same threshold as the primary NAAQS (Environmental Protection Agency, 2015b). Plants and ecosystems have been shown to be sensitive to lower $O_3$ concentrations, over longer-term exposures, and at different times of day and year than when NAAQS exceedances are frequent (e.g., Kurpius et al., 2002; Panek 2004; Panek and Ustin, 2005; Fares et al., 2013). While trends toward median $O_3$ levels observed at a large number of U.S. sites (Lefohn et al., 2017) may have decreased the number of NAAQS exceedances, benefits to plants and ecosystems

may be limited (Lefohn et al., 2018). The U.S. Environmental Protection Agency (EPA) has considered redefining the secondary standard to reflect ecological systems, with the W126 metric put forth (Environmental Protection Agency, 2010). W126 is a 12-h daily 3-month summation weighted to emphasize higher $O_3$ concentrations (Environmental Protection Agency, 2006; Environmental Protection Agency, 2016) that is used by the U.S. National Park Service. There are a number of other

concentration metrics used to quantify ecosystem $O_3$ impacts. In Europe, the AOT40 index is common, and is equal to all daytime (defined as solar radiation $\geq$ 50 W m$^{-2}$) hourly $O_3$ concentrations greater than 40 ppb. In the U.S., two widely used indices are the SUM0 and SUM06 (e.g., Panek et al., 2002), which are the sum of all daytime hourly $O_3$ mixing ratios greater than or equal to 0 ppb and 60 ppb, respectively. Similarly, the M12 metric is an average exposure metric computed over the same daily time window (8 am–8 pm LT) and representing the same hourly $O_3$ concentrations when computed over a 3-month

period as SUM0 (e.g., Tingey et al., 1991; Lefohn and Foley, 1993). For a global assessment of $O_3$ distribution and trends using a variety of ecosystem concentration metrics see Mills et al. (2018), which is part of the Total Ozone Assessment Report (TOAR).

    Even ecosystem-based concentration metrics are proxies of variable quality for $O_3$ impacts, if $O_3$ concentrations are not well-correlated with plant $O_3$ uptake (e.g., Emberson et al., 2000; Panek et al., 2002; Panek, 2004; Fares et al., 2010a). This is

because of temporal mismatches between when $O_3$ is high and when plants uptake $O_3$ from the atmosphere, with differences in high $O_3$ and efficient $O_3$ uptake occurring on both diurnal and seasonal timescales. While ecosystem $O_3$ impacts are best represented by direct measurements of the $O_3$ stomatal flux (e.g., Musselman et al., 2006; Fares et al., 2010a; Fares et al., 2010b), exceedances of flux-based standards are difficult to operationalize, as there are few long-term $O_3$ flux observational records and because reported thresholds, when available, are highly species-specific (Mills et al., 2011).

Under the 1977 Clean Air Act Amendments, selected national parks were designated as Class I Federal areas and, as part of this, the National Park Service began measuring $O_3$ concentrations in the 1980s, prioritizing national parks downwind of cities and polluted areas, including SNP (National Park Service, 2015b). Data from these monitors can be used to compute various $O_3$ concentration metrics; however, direct flux measurements do not exist in SNP, or other national parks, over long enough timescales to assess the effects of multi-year emissions controls. Forest survey data, which assess $O_3$ impacts by

monitoring changes in plants and forests from visible injury records and species population estimates, are limited, as they are labor- and time-intensive, requiring the evaluation of at least dozens of trees per stand to distinguish moderate levels of injury (Duriscoe et al., 1996). These studies occur at some time interval after exposure, making correlation to specific $O_3$ concentrations not possible. As a result, there is a need to assess trends using concentration metrics, but to do so with knowledge of when plant $O_3$ uptake is greatest.

In this paper, we report $O_3$ trends from 2001 to 2012 in SNP and the upwind SJV city of Visalia to study the effects of SJV emission controls on SNP $O_3$. We do not extend the analyses beyond 2012 as, beginning in late 2012, California experienced the worst drought in recorded history (Griffin and Anchukaitis, 2014; Diffenbaugh et al., 2015). Because $O_3$ concentrations are influenced by drought conditions (e.g., Jacob and Winner, 2009; Huang et al., 2016), we focus on the 2001 to 2012 time period. We compute trends in human health- and ecosystem-based concentration metrics separately when regional

environmental conditions favor plant $O_3$ uptake (springtime) and when high $O_3$ is most frequent ($O_3$ season). We describe these $O_3$ changes in Visalia and SNP as function of distance downwind of Visalia by way of data collected at two monitoring stations located on the western slope of the Sierra Nevada Mountains. We demonstrate the importance of transport of urban $NO_x$ from the SJV on trends in $O_3$ production ($PO_3$) chemistry in SNP. Finally, we discuss the descriptive power of various $O_3$ metrics and consider implications of a regulatory focus on human health-based standards to reduce ecosystem $O_3$ impacts in SNP.

## 2 Sequoia National Park (SNP) and the San Joaquin Valley (SJV)

SNP is located in the southern Sierra Nevada Mountains (Figure 1) and is part of the largest continuous wilderness in the contiguous U.S., which includes Kings Canyon NP and Yosemite NP. The SJV extends 250 miles in length and is situated between the Southern Coast Ranges to the west, the Sierra Nevada Mountains to the east, and the Tehachapi Mountains to the south. The southern SJV is the most productive agricultural region in the U.S., an oil and gas development area, and home to the cities of Fresno, Visalia, and Bakersfield. The same climatic conditions that support agriculture in the region, especially the numerous sunny days, are also favorable for $PO_3$. The high rates of local $PO_3$ (Pusede and Cohen, 2012; Pusede et al., 2014), diverse local emission sources outside historical regulatory focus, e.g., agricultural and energy development activities (e.g., Gentner et al., 2014a; Gentner et al., 2014b; Pusede and Cohen, 2012; Park et al., 2013), and surrounding mountain ranges that impede air flow out of the valley, have resulted in severe regional $O_3$ pollution. Four SJV cities rank among the ten most $O_3$-polluted cities in the U.S.: Bakersfield (ranked 2), Fresno (3), Visalia (4) and Modesto-Merced (6) (American Lung Association, 2016).

Multiple airflow patterns influence $O_3$ in SNP and the SJV (see Zhong et al. (2004) for a diagram). First, summertime (April–October) afternoon low-level winds in the southern SJV are generally from the west-northwest (represented by Visalia in Figure 2a). These winds are strengthened by an extended land-sea breeze, with onshore flow entering central California through the Carquinez Strait near the San Francisco Bay and diverging to the south into the SJV and north to the Sacramento Valley (e.g., Zaremba and Carroll, 1999; Dillon et al., 2002; Beaver and Palazoglu, 2009; Bianco et al., 2011). Second, at night, a recurring local flow pattern in the SJV, known as the Fresno eddy, recirculates air in the southern region of the valley around Bakersfield in the counterclockwise direction back to Fresno and Visalia, further enhancing $O_3$ pollution and precursors in these cities (e.g., Ewell et al., 1989; Beaver and Palazoglu, 2009). Third, the most populous and $O_3$-polluted cities in the southern SJV, Fresno, Visalia, and Bakersfield, are located along the eastern valley edge. Here, air movement is also affected by mountain-valley flow (e.g., Lamanna and Goldstein 1999; Zhong et al., 2004; Trousdell et al., 2016). During the day, thermally-driven upslope flow brings air from the valley floor to higher mountain elevations from the west-southwest (Figure 2). In Figure 3, a high elevation SNP site (Moro Rock, 36.5469 N, 118.7656 W, 2050 m ASL) is visibly above the SJV surface layer in the late morning, but within this polluted layer in late afternoon. At night, the direction of flow reverses and air moves downslope from the east-northeast (Figure 2). The prevalence of shallow nighttime surface inversions in the SJV means that

evening downslope valley flow at higher elevations may be stored within nocturnal residual layers and entrained into the surface layer the following morning.

## 3 Results

High $O_3$ days are most frequent in SNP and the SJV in the summer through early fall (Pusede and Cohen, 2012; Meyer and Esperanza, 2016), as $PO_3$ chemistry is often temperature-dependent (reviewed in Pusede et al., 2015) and this effect is particularly strong in the SJV (Pusede and Cohen, 2012; Pusede et al., 2014). The $O_3$ season is defined here as June–October with ~90% of annual $O_3$ 8-h NAAQS exceedances in SNP occur during $O_3$ season (2001–2012).

In the Sierra Nevada foothills, high rates of plant $O_3$ uptake are asynchronous with $O_3$ season because of the
Mediterranean climate (e.g., Kurpius et al., 2002; Kurpius et al., 2003; Panek, 2004). Plants also capture carbon dioxide required for photosynthesis and transpire through stomata; therefore, $O_3$ uptake is not only a function of the atmospheric $O_3$ concentration, but also of photosynthetically-active radiation (PAR), the inverse of the atmospheric vapour pressure deficit (VPD) (Kavassalis and Murphy, 2017), and soil moisture (e.g., Reich, 1987; Bauer et al., 2000; Fares et al., 2013). In SNP, PAR is highest in the late spring through early fall and VPD is at a minimum in winter and spring. In the Sierra Nevada
Mountains, plant water status (VPD and soil moisture) has been shown to explain up to 80% of day-to-day variability in stomatal conductance, with conductance decreasing with increasing water stress from mid-May to September and remaining low until soils are resaturated by wintertime precipitation. Plant $O_3$ uptake in Sierra Nevada forests has been reported to be greatest in April–May (Kurpius et al., 2002; Panek, 2004; Panek and Ustin, 2005).

In this context, we separately consider $O_3$ trends in springtime (April–May), which is when plant $O_3$ uptake best correlates
with variability in atmospheric $O_3$ concentrations in the region, and during $O_3$ season (June–October), which is when $O_3$ concentrations are highest. In this manuscript, for clarity we generally use the term *impacts* when discussing ecosystem metrics and *concentrations* when talking about human health metrics; $O_3$ ecosystem and human health effects are of course both $O_3$ impacts.

Hourly $O_3$ data have been routinely collected in SNP at two monitoring stations, a lower elevation site, SNP-Ash
Mountain (36.489 N, 118.829 W), at 515 m above sea level (ASL) and a higher elevation site, SNP-Lower Kaweah (36.566 N, 118.778 W), at 1926 m ASL (Figure 1). We refer to these stations as SEQ1 and SEQ2, respectively. $O_3$ and $NO_2$ data are measured in Visalia (36.333 N, 119.291 W), which is in the upwind direction of SNP at 102 m ASL (Figure 2). The data are collected by various agencies, including the National Park Service, and are hosted by the California Air Resources Board and available for download at https://www.arb.ca.gov/aqmis2/aqdselect.php.

### 3.1 Diurnal $O_3$ variability

Diurnal $O_3$ and $O_x$ ($O_x \equiv O_3 + NO_2$) concentrations are shown in Figure 4 in springtime (panel a) and $O_3$ season (panel b) over the 2001–2012 time period. Visalia data are shown as $O_x$ to account for the portion of $O_3$ stored as $NO_2$, which can be

substantial in the nearfield of fresh $NO_x$ emissions and at night. $NO_2$ data are not available in SEQ1 and SEQ2; however, these sites are removed from large $NO_x$ sources (Figure 1) and $O_3 \approx O_x$ is a reasonable approximation.

In Visalia, $O_x$ concentrations increase sharply beginning in early morning (7 am LT) until 2 pm LT, continuing to rise slightly until 4–5 pm LT (Figure 4). This diurnal pattern reflects a combination of local $PO_3$ (the initial rise) and advection of $O_x$ from the upwind source region (late afternoon maximum). In the morning, enhanced rush-hour $NO_x$ emissions overlap in time with the initial increase in $O_x$ with 30–40% of $O_x$ as $NO_2$ at 7–8 am LT. In the afternoon, from 12–4 pm LT ~10–15% of $O_x$ is $NO_2$. At 5 pm, $NO_2$ concentrations increase with evening rush hour with 30–40% of $O_x$ as $NO_2$ at 5–6 pm LT.

Diurnal $O_3$ variability at SEQ1 and SEQ2 is characterized by two features, an early morning rise (6 am LT) and an increase in the late afternoon (3–4 pm LT). The timing of this morning $O_3$ increase is consistent with entrainment of $O_3$ in nocturnal residual layers aloft during morning boundary layer growth. The influence is substantial, as morning $O_3$ accounts for 50% (springtime and $O_3$ season) of the daily change in $O_3$ at SEQ1 and 50% (springtime) and 37% ($O_3$ season) of the daily change in $O_3$ at SEQ2. The timing of afternoon peak $O_3$ is consistent with upslope air transport from the SJV (Figure 2). If $O_3$ attributed to local $PO_3$ in Visalia is greatest around 2 pm LT, typical of many urban locations, with mean winds at SEQ1 of 3 m s$^{-1}$ and SEQ2 of 2 m s$^{-1}$, we expect $O_3$ to peak in SEQ1 at ~5 pm (45 km downwind of Visalia) and at SEQ2 shortly after (9.7 km downwind of SEQ1, which includes the change in elevation using the Pythagorean theorem). This is broadly what we observe. While the actual distance of airflow is dictated by the mountain terrain and a parcel of air will travel a distance longer than the straight-line path on a smooth surface, the timing of the $O_3$ diurnal patterns is consistent with airflow travel time roughly equal to that determined by the horizontal distance and mean wind speed. There has been no change in the hour of peak $O_3$ mixing ratio at either SEQ1 or SEQ2 over the 2001 to 2012 period.

**3.2 Weekday-weekend $O_3$ variability**

SNP and the SJV are in close geographic proximity but their local $PO_3$ regimes are different. In 2016, as part of the Korea-U.S. Air Quality (KORUS-AQ) experiment (https://www.air.larc.nasa.gov/missions/korus-aq/index.html) and Student Airborne Research Program (SARP), the NASA DC-8 sampled a low-altitude transect (~130 m above ground level) along the trajectory of SJV mountain-valley outflow. The DC-8 flew at ~10 am LT from Orange Cove, an SJV town 35 km north of Visalia, 24 km up the western slope of the Sierra Nevada Mountains to an elevation of ~1000 m ASL. In Figure 5, the change in $NO_x$ and isoprene along this transect is shown as a function of change in surface elevation. Boundary layer $NO_x$ is observed to decrease with increasing distance downwind of the SJV, while isoprene concentrations increase. Isoprene is a large source of reactivity in the Sierra Nevada foothills (e.g., Beaver et al., 2012; Dreyfus et al., 2002) and the combined $NO_x$ and isoprene gradients suggest potentially distinct $PO_3$ regimes in the SJV and SNP. While these data were collected on one day in a different year from our study, the relative pattern of $NO_x$ to organic compound emissions is likely representative, as there have been no substantial changes in the locations of urban $NO_x$ and biogenic organic emitters. This $NO_x$ to organic compound gradient is consistent with observations over longer sampling periods downwind of the Central California city of Sacramento, where the

NO$_x$-enriched Sacramento urban plume is transported up the western slope of the vegetated Sierra Nevada Mountains (e.g., Beaver et al., 2012; Dillion et la., 2002; Murphy et al., 2006).

If the major source of O$_3$ in SNP is O$_3$ produced in the SJV and transported downwind, then the observed NO$_x$ dependence of $PO_3$ in SNP and the SJV would be the same even if $PO_3$ regimes in the two locations were different. To test this hypothesis, we consider O$_3$ in SNP and O$_x$ in the SJV separately on weekdays and weekends. Weekday-weekend NO$_x$ concentration differences are well-documented across the U.S. (e.g., Russell et al., 2012) and California (e.g., Marr and Harley, 2002; Russell et al., 2010), and are caused by reduced weekend heavy-duty diesel truck traffic, where heavy-duty diesel trucks are large sources of NO$_x$ but not O$_3$-forming organic gases. As a result, NO$_x$ concentrations are typically 30–60% lower on weekends than weekdays and these NO$_x$ changes occur without comparably large decreases in reactive organic compounds (e.g., Pusede et al., 2014). $PO_3$ is the only term in the O$_3$ derivative expected to exhibit NO$_x$ dependence.

We focus on the earliest 3-yr time period in our record, 2001–2003, which is when differences in $PO_3$ chemical sensitivity in the SJV and SNP are expected to be most pronounced (Pusede and Cohen, 2012). We define weekdays as Tuesdays–Fridays and weekends as Sundays to avoid atmospheric memory effects. Statistics were sufficient to minimize any co-occurring variation in meteorology, with no significant weekday-weekend differences observed in daily maximum temperature, wind speed, or wind direction. We focus on afternoon (12–6 pm LT) O$_x$, when O$_3$ concentrations in SNP are most influenced by the SJV (from Figure 4). We also compare weekday-weekend O$_x$ at high and moderate temperatures, with temperature regimes defined as days above and below the 2001–2012 seasonal mean daily maximum average temperature in Visalia. Temperatures in Visalia are well correlated ($R^2 = 0.98$) with temperatures in SEQ1 over 2001–2012. During springtime and O$_3$ season, mean maximum average temperatures in Visalia were 25.1 ± 5.9 and 32.0 ± 5.3 °C (ranges are 1σ variability), respectively.

At moderate temperatures, statistically significant weekday-weekend differences were observed (Table 1). During O$_3$ season, O$_x$ was 6.3 ± 3.5% higher on weekends than weekdays (relative to weekdays) in Visalia, indicating local $PO_3$ was NO$_x$ suppressed. At the same time, O$_3$ was 4.6 ± 3.3% and 4.9 ± 3.9% higher on weekdays than weekends at SEQ1 and SEQ2, respectively, implying $PO_3$ in SNP was NO$_x$ limited. A similar pattern was observed during springtime, as O$_x$ was 7.4 ± 4.6% higher on weekends than weekdays in Visalia and O$_3$ was 3.5 ± 7.4% and 4.7 ± 5.5% higher on weekdays than weekends in SEQ1 and SEQ2. These weekday-weekend patterns imply that a substantial portion of O$_3$ in SNP is produced by low-NO$_x$ $PO_3$ chemistry during air transport from the SJV. At high temperatures, greater weekday concentrations in O$_x$ in Visalia and O$_3$ at SEQ1 and SEQ2 imply NO$_x$-limited chemistry in all three locations (Table 1). Averaged across sites, percent differences in weekdays and weekends were 8.7 ± 4.8% in the springtime and 4.3 ± 2.3% during O$_3$ season. $PO_3$ during upslope transport is not apparent by this method because O$_3$-season $PO_3$ was also NO$_x$ limited in Visalia, indicating a portion of O$_3$-forminig organic reactivity in Visalia was temperature dependent, consistent with past analyses in other SJV cities (Steiner et al., 2006; Pusede and Cohen, 2012; Pusede et al., 2014; Rasmussen et al., 2014).

### 3.3 O$_3$ trends over time

In Figure 6 and Table 2, we report 12-yr $O_3$ trends (2001–2012) in SNP and the SJV in springtime and during $O_3$ season using four concentration metrics: MD8A; two common vegetative-based indices, SUM0 and W126; and a morning average metric. We do not report trends in SUM06 or AOT40 vegetative indices, as they have been shown to poorly correlate with $O_3$ uptake at a Sierra Nevada forest site even in springtime (Panek et al., 2002; Fares et al., 2010b). MD8A $O_3$ is a human health-
based metric computed as the maximum unweighted daily 8-h average $O_3$ mixing ratio. A region is classified as in nonattainment of the NAAQS when the fourth-highest MD8A $O_3$ over a 3-yr period, known as the design value, exceeds a given standard. In this work, we utilize the seasonal mean MD8A and discuss $O_3$ exceedances as individual days in which MD8A $O_3 > 70.9$ ppb, the current 8-h NAAQS (CFR 40, 2015). SUM0 is equal to the sum of hourly $O_3$ concentrations over a 12-h daylight period (8 am–8 pm LT), as opposed to SUM06, which is limited to hourly $O_3$ mixing ratios greater than 60 ppb.
SUM0 is based on the assumption that the total $O_3$ dose has a greater impact on plants than shorter duration high $O_3$ exposures (Kurpius et al., 2002). The summation is unweighted, attributing equal significance to high and low $O_3$ concentrations (Musselman et al., 2006). SUM0 averaging is restricted to time periods when stomata are open (daylight), a condition not required for the MD8A. W126 is a weighted summation (8 am–8 pm LT), assuming higher $O_3$ is more damaging to plants than lower $O_3$ levels. W126 weighting is sigmoidal, with hourly $O_3$ weights equal to $(1 + 4403e^{-126[O3]})^{-1}$, such that hourly mixing
ratios below (above) 60 ppb receive less (more) weight (Environmental Protection Agency, 2016). Here, SUM0 and W126 summations are computed following the W126 protocol (Environmental Protection Agency, 2016), affording straightforward comparisons between the metrics. First, in months with less than 75% of hourly data coverage in the 8 am–8 pm LT window, missing values are replaced with the lowest observed hourly measurement over the study period (April–October) only until the dataset is 75% complete. From 2001–2012, 0, 8, and 3 months were initially less than 75% complete in Visalia, SEQ1, and
SEQ2, respectively. Second, monthly summations of daily indices, comprised of hourly data (8 am–7 pm), are computed; when data are missing, the summation is divided by the data completeness fraction. Consecutive 3-month metrics are computed by adding monthly indices. In practice, SUM0 and W126 are computed as 3-yr averages of the highest 3-month summation; however, we define springtime SUM0 and W126 as the 3-month summation over April–June and $O_3$ season SUM0 and W126 as the mean of the 3-month summations over June–August, July–September, and August–October (not the highest of the three
3-month sums). Because less than 15% of data were available for August 2008 at SEQ1, $O_3$ season SUM0 and W126 were computed as the mean of 3-month summations over June, July, and September, and July, September, and October only for this site and year. We compute morning (7 am–12 pm LT) trends ($O_x$ in Visalia and $O_3$ in SNP), as high $O_3$ plant uptake rates (in the morning) and high $O_3$ concentrations (in the afternoon) are out of phase within daily timeframes in the Sierra Nevada Mountains. Plant $O_3$ uptake typically follows a pattern of rapid morning uptake, relatively constant flux through midday, and
a decrease in uptake in afternoon as plants close their stomata to prevent water loss in the hot, dry afternoon (Kurpius et al., 2002; Fares et al., 2013). Efficient morning uptake occurs because plants recharge their water supply overnight, which with low morning temperatures and VPD, results in high stomatal conductance (Bauer et al., 2000). Morning uptake in the Sierra Nevada maximizes in springtime around 8 am LT (Kurpius et al., 2002; Panek and Ustin, 2005; Fares et al., 2013).

In Figure 6, mean seasonal daily MD8A and morning metrics and cumulative SUM0 and W126 metrics are shown for Visalia, SEQ1, and SEQ2 with their fit derived using an ordinary least squares linear regression. Table 2 reports both the regression slope value (right columns) and the change in $O_3$ relative to the $O_3$ season fit value in SEQ1 in 2001 reported as a percent (left columns). SEQ1 experiences the highest $O_3$ observed for each metric and using a standard denominator facilitates comparison between monitoring sites and between seasons. Table 2 coloration indicates trend significance computed using the Mann-Kendall non-parametric test following the categorization developed by TOAR authors (Chang et al., 2017; Mills et al., 2018; Lefohn et al., 2018), with p-values deemed statistically significant (0–0.05), indicative of a trend (0.05–0.10), weakly indicative of change (0.10–0.34), and indicative of weak or no change (0.34–1).

Three patterns emerge in SNP $O_3$ trends over time: (1) $O_3$ decreased everywhere over the 12-yr record by all metrics in both seasons; (2) $O_3$ decreased at a slower rate in the springtime than during $O_3$ season by most metrics; and (3) $O_3$ decreased more rapidly in SNP versus Visalia and at SEQ2 versus SEQ1.

Seasonal differences in $O_3$ trends are prominent at each site. For example, $O_3$ at SEQ1 generally decreased less in springtime than during $O_3$ season (Table 2). For context in SEQ1, during $O_3$ season the mean MD8A declined from 82.3 ppb (2001–2002) to 73.8 ppb (2011–2012), but in the springtime the MD8A fell from 61.7 ppb (2001–2002) to 55.6 ppb (2011–2012). SUM0 $O_3$ fell from 87.0 ppm h (2001–2002) to 79.0 ppm h (2011–2012) during $O_3$ season and from 69.9 ppm h (2001–2002) to 61.8 ppm h (2011–2012) in the springtime. W126 $O_3$ decreased from 67.8 ppm h (2001–2002) to 53.7 ppm h (2011–2012) during $O_3$ season and from 39.8 ppm h (2001–2002) to 25.4 ppm h (2011–2012) in springtime. Morning $O_3$ fell from 67.1 ppb (2001–2002) to 59.6 ppb (2011–2012) during $O_3$ season and from 49.0 ppb (2001–2002) to 45.1 ppb (2011–2012). This pattern was not observed in one instance: SUM0 in SEQ2. Here, seasonal differences were comparable; however, mean daily indices were observed to differ, where SUM0 $O_3$ decreased from 0.914 ppm h (2001–2002) to 0.816 ppm h (2011–2011) during $O_3$ season, and, in the springtime, fell from 0.673 ppm h (2001–2002) to 0.616 ppm h (2011–2012), which amount to a change of –11% during $O_3$ season and –8% in the springtime.

Additionally, greater $O_3$ decreases were observed at SEQ1 than Visalia and at SEQ2 compared to SEQ1. Over the 12-yr period, MD8A $O_3$ declined at a rate of 46% ($O_3$ season) and 61% (springtime) faster at SEQ1 than in Visalia, and 29% ($O_3$ season) and 41% (springtime) faster at SEQ2 than SEQ1 (based on the slopes reported in Table 2). SUM0 and W126 $O_3$ decreased 79% and 59% ($O_3$ season) and 38% and 54% (springtime) faster at SEQ1 than in Visalia and 20% and 23% ($O_3$ season) and 58% and 17% (springtime) faster at SEQ2 than SEQ1. Morning $O_x$ trends at SEQ1 and Visalia were similar in springtime, but $O_x$ decreased 40% more rapidly at SEQ1 during $O_3$ season and faster at SEQ2 than SEQ1 by 17% ($O_3$ season) and 55% (springtime). For each metric, we observe greater interannual variability relative to the net decline in springtime than during $O_3$ season. This site-dependence is reflected in the $O_3$ trend p-values (Table 2), where at SEQ2, slopes are either statistically significant at the 0.05 level or indicative of a trend (0.05–0.10) for each metric in both seasons. At SEQ1, slopes are statistically significant during $O_3$ season, but only indicative (W126), weakly indicative (MD8A), or suggestive of weak to no change (SUM0 and Morning $O_x$) in springtime. Trends in Visalia are the least robust, with p-values typically only weakly indicative or suggestive of minor to no change in both springtime and $O_3$ season.

High O$_3$, as defined by exceedances of protective thresholds, also became less frequent over the 12-yr record. The number of days in which MD8A O$_3$ was greater than 70.9 ppb in 2001–2002 (averages are rounded up) was 66 yr$^{-1}$ (O$_3$ season) and 14 yr$^{-1}$ (springtime) in Visalia. In 2011–2012, the number of exceedances fell to 39 yr$^{-1}$ (O$_3$ season) and 6 yr$^{-1}$ (springtime). At SEQ1 in 2001–2002, there were 119 exceedance days yr$^{-1}$ (O$_3$ season) and 20 yr$^{-1}$ (springtime), declining in 2011–2012 to 97 yr$^{-1}$ (O$_3$ season) and 10 yr$^{-1}$ (springtime). At SEQ2 in 2001–2002, there were 103 exceedance days yr$^{-1}$ (O$_3$ season) and 13 yr$^{-1}$ (springtime). In 2011–2012, this decreased to 62 exceedance days yr$^{-1}$ (O$_3$ season) and 3 yr$^{-1}$ in 2011–2012 (springtime). Patterns in high MD8A O$_3$ days follow trends in other metrics, with the largest rates of change occurring during O$_3$ season in SEQ2 (–4.7 days yr$^{-1}$, $p = 0.02$), then SEQ1 (–2.8 days yr$^{-1}$, $p = 0.05$) and Visalia (–2.5 days yr$^{-1}$, $p = 0.05$). In springtime, smaller decreases are observed with similar spatial patterns, SEQ2 (–1.0 days yr$^{-1}$, $p = 0.02$), SEQ1 (–1.0 days yr$^{-1}$, $p = 0.15$) and Visalia (–0.5 days yr$^{-1}$, $p = 0.37$).

While there is no standard for SUM0, there are three time-integrated W126 protective thresholds. These are: 5–9 ppm h to protect against visible foliar injury to natural ecosystems, 7–13 ppm h to protect against growth effects to tree seedlings in natural forest stands, and 9–14 ppm h to protect against growth effects to tree seedlings in plantations, known as the 5, 7, and 9 ppm h standards (Heck and Cowling 1997). The EPA has considered a potential secondary W126 ozone standard between 7 and 17 ppm h (Environmental Protection Agency, 2015a); likewise, the Clean Air Science Advisory Committee recommended a W126 standard level between 7 and 15 ppm h (Environmental Protection Agency, 2015a). In this work, rather than calculate W126 exceedances using a 3-month summation of monthly indices, we instead count the number of days required for an exceedance to occur, summing daily W126 indices from the first day of the springtime (1 April). A larger number of days indicates improved air quality. We do this to generate information in addition to exceedance frequency, as W126 O$_3$ at SEQ1 and SEQ2 is greater than all three standards in all years in both seasons. We only consider springtime, as this is when W126 is reported to better correlate with plant O$_3$ uptake (Panek et al., 2002; Kurpius et al., 2002; Bauer et al., 2000). At SEQ1 from 1 April in 2001–2002, 37, 41, and 45 days of O$_3$ accumulation reached exceedances of the 5, 7, and 9 ppm h thresholds, respectively (averages are rounded up). In 2011–2012, 3 to 13 more days were needed at SEQ1, as 40, 49, and 58 days of O$_3$ accumulation were required to exceed the 5, 7, and 9 ppm h thresholds. At SEQ2 from 1 April in 2001–2002, 41, 46, and 49 days of accumulation led to exceedance of the 5, 7, and 9 ppm h thresholds, respectively. In 2011–2012, 59, 65, and 73 days were required at SEQ2, or 18–24 more days.

## 4 Discussion

### 4.1 O$_3$ metrics

Long-term measurements of O$_3$ fluxes rather than O$_3$ concentrations are required to fully understand the effects of upwind emission controls on ecosystem O$_3$ impacts. This is particularly true in Mediterranean ecosystems like SNP and under drought conditions, which is where and when plant O$_3$ uptake and high atmospheric O$_3$ concentrations may be uncorrelated (e.g., Panek

et al., 2002). This may also be true in European Mediterranean climate regions, where high concentrations of ecosystem-based $O_3$ metrics have also been observed (Mills et al., 2018). We have based our analysis on results from years of $O_3$ flux data collected in forests on the western slope of the Sierra Nevada Mountains (Bauer et al., 2000; Panek and Goldstein, 2001; Panek et al., 2002; Kurpius et al., 2002; Fares et al., 2010; Fares et al., 2013); however, there are few other $O_3$ flux datasets that span

multiyear timescales and no flux observations in SNP. In California, flux measurements suggest springtime SUM0 trends offer the most insight into trends in ecosystem $O_3$ impacts in SNP; that said, we find similar conclusions would be drawn regarding multiyear $O_3$ variability by location by assessing trends in SUM0, MD8A $O_3$, and the morning $O_x$ metric. This can be explained by the upslope-downslope air flow in our study region and is evident in SNP diurnal $O_3$ patterns (Figure 4), which show considerable $O_3$ entrained into the boundary layer in the morning. $O_3$ concentrations are strongly influenced by afternoon

concentrations on the previous day. Comparable trends in morning, afternoon, and daily average $O_3$ would then arise under conditions of persistence, which are common in Central California, but these results may not extend to other downwind ecosystems in the absence of an upslope-downslope flow pattern. The dynamically-driven elevated morning $O_3$ concentrations have important consequences for plants, as vegetation in SNP may be particularly vulnerable because plant $O_3$ uptake rates are often highest in the morning.

Reductions in ecosystem $O_3$ impacts as represented by declines in W126 are greater than those of SUM0. We attribute this difference to the W126 weighting algorithm that makes the metric most sensitive to changes in the highest $O_3$. Using the GEOS-Chem model with a focus on national parks, Lapina et al. (2014) also found W126 was more responsive to decreases in anthropogenic emissions than daily (8 am–7 pm, LT) average $O_3$ concentrations. With the Community Earth System Model, Val Martin et al. (2015) modeled air quality in national parks under two Representative Concentration Pathway (RCP)

scenarios, computing substantially larger decreases over a 50-yr period in W126 $O_3$ compared to the MD8A. In the TOAR global analysis, Mills et al. (2018) found April–September W126 downward trends over 1995–2014 in California of between 1–2 ppm h $yr^{-1}$ to be among the most rapid W126 declines in the world. Considering that the SUM0 metric has been shown to best correspond to plant $O_3$ uptake in Sierra Nevada forests using $O_3$ flux observations (Panek et al., 2002) and that we observe W126 $O_3$ has declined at approximately twice the rate of SUM0 over 2001–2012, W126 trends may provide an overly

optimistic representation of past declines in ecosystem $O_3$ impacts in SNP.

## 4.2 Reducing high $O_3$ in SNP and polluted downwind ecosystems

$NO_x$ decreases have generally made greater improvements in $O_3$ in SEQ1 than Visalia and in SEQ2 than SEQ1, a trend that corresponds to increasing distance downwind of the SJV. We attribute this to the importance of export of $NO_x$ from the SJV on $O_3$ in SNP, combined with distinct $PO_3$ chemical regimes in SNP versus Visalia. Evidence for this is four-fold. First, $O_3$ at

SEQ1 is greater than $O_x$ in Visalia, at least during $O_3$ season, suggesting net $O_3$ formation as air travels from the SJV to SNP. Second, according to observations of $O_x$ (Visalia) and $O_3$ (SNP) on weekdays versus weekends, $PO_3$ was simultaneously $NO_x$-suppressed in Visalia and $NO_x$-limited in SNP, with the weekday-weekend dependence of $O_3$ reflecting the chemical regime in which it is produced. Third, aircraft observations collected in the direction of daytime upslope flow from the SJV to Sierra

Nevada foothills reveal substantial decreases in $NO_x$ concentrations relative to isoprene, a key contributor to total organic reactivity (e.g., Beaver et al., 2012). Fourth, $O_3$ decreases (2001–2012) are observed to be greater in SNP than Visalia, and greater with increasing distance downwind. Distinct local $PO_3$ regimes lead to $PO_3$ chemistry in Visalia and SNP that is differently sensitive to emission controls, with $NO_x$-limited SNP historically more responsive to $NO_x$ emission control than

Visalia. SNP $NO_x$-limitation is enhanced by $NO_x$ dilution during transport, which further decreases $NO_x$ relative to the abundance of local organic compounds. Downwind sites usually experience $PO_3$ chemistry that is more $NO_x$-limited than in the often $NO_x$-suppressed (or at least more $NO_x$-suppressed) urban core. As a result, we expect similar location-specific $O_3$ trends in other ecosystems and national parks downwind of major $NO_x$ sources like cities. However, while the extent of observed $O_3$ improvements in SNP follows the pattern of increasing distance downwind of Visalia with sustained $NO_x$ emission

control in the SJV (Russell et al., 2010; Pusede and Cohen, 2012), $PO_3$ chemistry is non-linear and the direction of location-specific trends may vary. That said, at some distance downwind this conclusion breaks down, as areas become less and less influenced by the upwind source.

Because $PO_3$ in SNP is $NO_x$-limited, future $NO_x$ reductions are expected to have at least as large an impact on local $PO_3$ as past reductions. Seasonal mean $NO_2$ concentrations have decreased by 58% and 53% in Visalia in springtime and $O_3$ season

over our study window, respectively. Local $NO_x$ emissions should continue to decline into the future, as there are significant controls currently ongoing or in the implementation phase, including more stringent national rules on heavy-duty diesel engines (Environmental Protection Agency, 2000; 2010), combined with California Air Resources Board (CARB) diesel engine retrofit-replacement requirements (California Air Resources Board, 2008; 2014), and more stringent CARB standards for gasoline-powered vehicles (California Air Resources Board, 2012). While $O_3$ declines near or greater than those that occurred

from 2001 to 2012 are required to eliminate exceedances in SNP, modeling analysis by Lapina et al. (2014) suggests that W126 in the region would be well below these thresholds in the absence of anthropogenic precursor emissions, implying further emissions controls would be effective. Under the stringent precursor controls of RCP4.5, Val Martin et al. (2015) projected decreases of 11% and 67% for the MD8A and W126 in 2050, respectively, from the base year of 2000, with mean $O_3$ decreasing from 58.9 ppb (MD8A) and 45.5 ppm h (W126) in 2000 to 52.7 ppb (MD8A) and 15.1 ppm h (W126). Under the RCP8.5,

smaller $O_3$ declines were predicted, with MD8A unchanged and W126 falling by 38% to 28.3 ppm h. Given that these scenarios represent a reasonable spread of possible future climatic conditions, Val Martin et al. (2015) suggest at least W126 will remain well above protective thresholds in 2050.

Over 2001–2012, $O_3$ declines have mostly been smaller in SNP when plant $O_3$ uptake is greatest (springtime), despite comparable $NO_x$ decreases in both seasons. This may be in part because regulatory strategies prioritize attainment of the $O_3$

NAAQS in polluted urban areas like the SJV basin, where air parcels influenced by the results of these controls are then transported downwind to locations with different $PO_3$ chemistry. In the development of regulatory plans, agencies use models to hindcast past $O_3$ episodes, facilitating testing of the efficacy of specific $NO_x$ and/or organic emissions reductions over that episode to meet the 8-h $O_3$ NAAQS or progress goals (Environmental Protection Agency, 2007; Environmental Protection Agency, 2014). In nonattainment areas, U.S. EPA guidance recommends modeling past time periods that meet a number of

specific criteria, such as typifying the meteorological conditions that correspond to high $O_3$ days as defined by the MD8A greater than the NAAQS value and focusing on the ten highest modeled $O_3$ days (Environmental Protection Agency, 2007; Environmental Protection Agency, 2014). Regulatory modeling in the SJV (Visalia, SEQ1, and SEQ2 are included in this attainment demonstration) is more comprehensive, as it was recently updated to span the full $O_3$ season (defined as May–

September); still potential reductions (known as relative reduction factors, RRFs) are based on the MD8A and restricted to high $O_3$ days (San Joaquin Valley Air Pollution Control District, 2007; San Joaquin Valley Air Pollution Control District, 2014). In the SJV, high $O_3$ days are most frequent in the late summer ($O_3$ season) and on the hottest days of the year (Pusede and Cohen, 2012). Even in SEQ1 and SEQ2, days with MD8A > 70.9 ppb are far more common in the summer. Because of chemical and meteorological differences between seasons, this may lead to policies not optimized to decrease $O_3$ in cooler

springtime conditions, which in the SJV are more $NO_x$-suppressed and therefore more sensitive to controls on reactive organic compounds (Pusede et al., 2014). In addition, we observe greater year-to-year $O_3$ variability in the springtime than during $O_3$ season (Figure 6), suggestive of a larger relative role of interannual meteorological variability controlling $O_3$ concentrations. Deeper cuts in emissions appear to be required in the springtime in SNP, as decreases in anthropogenic emissions have a smaller effect, both relatively and in the absolute, on the total $O_3$ abundance than during $O_3$ season, in part because background

$O_3$ makes the greatest contribution to daily $O_3$ in the springtime SNP (Figure 4).

The contribution of background $O_3$ concentrations and non-local sources challenges regulators (Cooper et al., 2015), as natural sources produce $O_3$ even in the absence of anthropogenic precursor emissions, $O_3$ can be transported over significant distances, and $O_3$ concentrations are influenced by large-scale meteorological and climatic events. Multiple studies have identified an increasing trend in $O_3$ at rural sites (often used as a proxy for background $O_3$) in the western U.S., particularly in

the springtime (e.g., Cooper et al., 2012, Lin et al., 2017). Parrish et al. (2017) presented observational evidence of a slowdown and reversal of this trend on the California west coast since 2000, though the reversal was stronger in the summer than springtime. Using observations and the GFDL-AM3 model, Lin et al. (2017) computed that Asian anthropogenic emissions accounted for 50% of simulated springtime $O_3$ increases at western U.S. rural sites, followed by rising global methane (13%) and variability in biomass burning (6%). Northern mid-latitude transport of Asian pollution to the western U.S. is strongest

during March–April and weakest in the summertime (e.g., Wild and Akimoto, 2001; Liu et al., 2003; Liu et al., 2005), with high-elevation locations in the Sierra Nevada Mountains being more vulnerable to reception of Asian $O_3$ and $O_3$ precursors (e.g., Vicars and Sickman, 2001; Heald et al., 2003; Hudman et al., 2004). Hudman et al. (2004) compared surface observations with GEOS-Chem-modeled $O_3$ enhancements in Asian pollution outflow, finding that, on average, transport events in April– May 2002 led to $8 \pm 2$ ppb higher MD8A $O_3$ concentrations at SEQ2. East Asian $NO_x$ emissions have risen over our study

window (e.g., Miyazaki et al., 2017), potentially causing an increase in the influence of trans-Pacific transport on $O_3$ concentrations at SEQ2 and reducing the efficacy of local $NO_x$ control in springtime. However, $NO_x$ emission and concentration declines have been observed over China since 2011 (Liu et al., 2016), diminishing possible influence of Asian transport events in SNP. Background $O_3$ concentrations are also responsive to large-scale climatic events, and elevated springtime $O_3$ at rural sites in the western U.S. has been linked to strong La Niña winters (Lin et al., 2015; Xu et al., 2017),

which are associated with an increased frequency of deep tropopause folds that entrain $O_3$-rich stratospheric air into the troposphere (Lin et al., 2015). Over our study period, strong La Niña events occurred during the winter of 2007–2008 and 2010–2011. In general, transport of Asian pollution and tropopause folds are expected to have a greater impact in the springtime and at the higher-elevation SEQ2. While we do observe smaller decreases in $O_3$ in springtime at SEQ2 than during $O_3$ season, interannual trends have been more downward at SEQ2 than at the lower elevation sites, SEQ1 and Visalia, in both seasons. This suggests that these factors may impact surface $O_3$ at high-elevations in SNP during individual events (e.g., Hudman et al., 2004) but that interannual trends in seasonal averages are more influenced by chemistry during upslope outflow from the SJV.

## 5 Conclusions

We describe $O_3$ trends at two monitoring stations in SNP and in the SJV city of Visalia, which is located in the upwind direction from SNP. We show that a major portion of the $O_3$ concentration in SNP is formed during transport from $NO_x$ emitted in the SJV, rather than from $O_3$ produced in Visalia and subsequently transported downwind. This has contributed to reductions in $O_3$ in SNP over the 12-yr period of 2001–2012, even while $PO_3$ in Visalia was $NO_x$ suppressed. Evidence for this includes greater $O_3$ at SEQ1 than $O_x$ in Visalia during $O_3$ season (Figure 4), distinct weekday-weekend $O_3$ differences in SNP and Visalia (Table 1), steep gradients in $NO_x$ and isoprene measured in the direction of upslope airflow out of the SJV within the boundary layer (Figure 5), and larger $O_3$ decreases over 2001–2012 at SEQ1 versus Visalia and at SEQ2 versus SEQ1 (Figure 6, Table 2).

We compute interannual $O_3$ trends using human health- and ecosystem-based concentration metrics in springtime and $O_3$ season separately in order to distinguish between ecosystem $O_3$ impacts (plant $O_3$ uptake) and high $O_3$ concentrations. We find that $O_3$ has decreased in SNP and Visalia by all metrics in both seasons consistent with ongoing $NO_x$ emission controls but observe smaller $O_3$ declines in springtime when plant uptake is greatest. The three metrics, MD8A, SUM0, and morning $O_x$, all indicate comparable reductions in $O_3$ over 2001–2012, with decreases of ~7% (springtime) and ~13% ($O_3$ season) at SEQ1 and 13–16% (springtime) and 15–19% ($O_3$ season) at SEQ2. We attribute similarity across these three metrics to upslope-downslope airflow at the eastern edge of the SJV, as morning $O_x$ and SUM0 are strongly affected by high afternoon $O_3$ concentrations on the previous day which results from the mixing of $O_3$-polluted nocturnal residual layers into the surface boundary layer. Past $O_3$ flux measurements in the region indicate the highest plant $O_3$ uptake in the springtime morning, therefore SNP vegetation experiences greater $O_3$ exposure than in locations without this memory effect. $O_3$ decreases over 2001–2012 computed with W126 are almost double those for SUM0, with the W126 emphasis of higher $O_3$ concentrations giving the most optimistic evaluation of the efficacy of past emission controls.

Diurnal and seasonal mismatches between plant $O_3$ uptake rates and $O_3$ concentration-based metrics make it challenging to accurately assess vegetative $O_3$ damage and to quantitatively evaluate the success of regulatory action on ecosystems. Future work would benefit from the development of an environmentally- and biologically-relevant metric that captures patterns in

plant $O_3$ uptake over daily and seasonal timescales, especially in Mediterranean ecosystems, where conditions conducive to plant $O_3$ uptake are asynchronous with conditions that lead to high $O_3$ concentrations.

**Acknowledgements**

Funding was provided by the NASA Student Airborne Research Program (SARP), National Suborbital Education and Research Center (NSERC), and the NASA Airborne Science Program (ASP). SEP was supported by NASA grant NNX16AC17G. We thank Philipp Eichler, Tomas Mikoviny and Armin Wisthaler for providing the DC-8 isoprene data. Isoprene measurements during KORUS-AQ were supported by the Austrian Federal Ministry for Transport, Innovation and Technology (bmvit) through the Austrian Space Applications Programme (ASAP) of the Austrian Research Promotion Agency (FFG). We thank Andrew Weinheimer at the National Center for Atmospheric Research for the providing the DC-8 NO and $NO_2$ measurements. We thank the pilots and crew of the NASA DC-8 and the KORUS-AQ science team. We acknowledge the California Air Resources Board for use of publicly-available $O_3$, $NO_2$, wind, and temperature measurements.

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

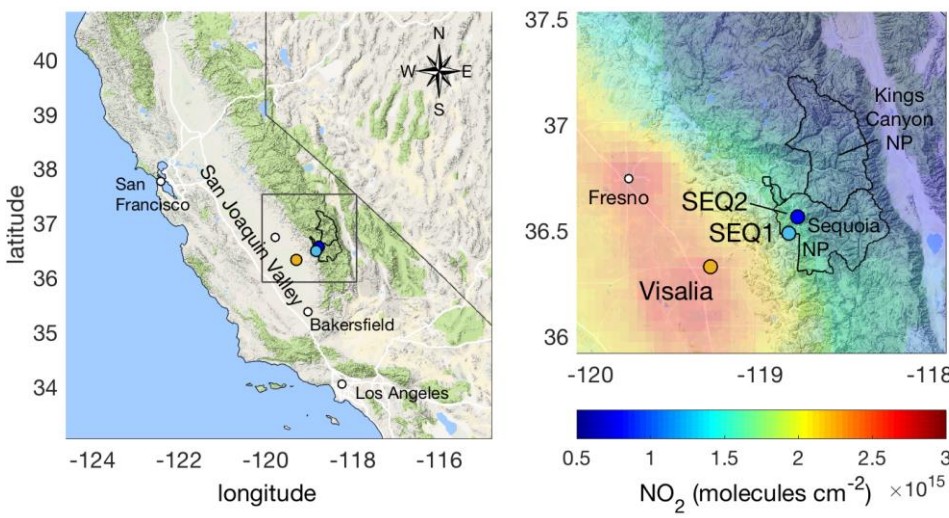

**Figure 1.** Map of California **(left)** with study region detail **(right)** indicating the locations of the SJV station, Visalia (orange), and two monitoring sites in SNP, SEQ1 (cyan) and SEQ2 (dark blue), with mean April–October, 2010–2012 OMI NO2 columns using the BEHR (Berkeley High-Resolution) product (Russell et al., 2011).

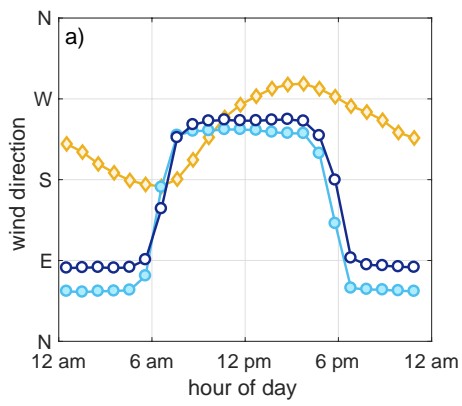

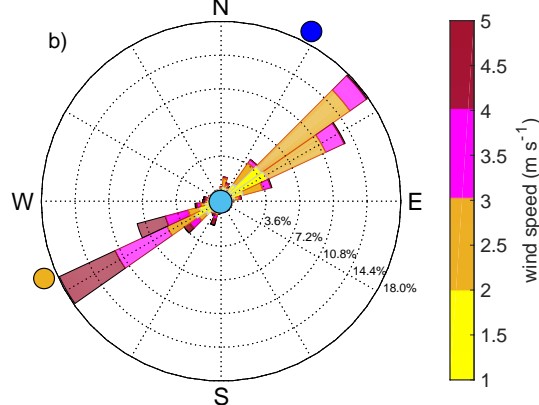

**Figure 2.** Hourly mean wind directions in Visalia (orange diamonds), SEQ1 (cyan filled circles), and SEQ2 (dark blue open circles) in April–October, 2001–2012 (panel a). Wind rose for SEQ1 (panel b) with the direction of the neighboring sites of Visalia (orange), SEQ1 (cyan), and SEQ2 (dark blue) indicated.

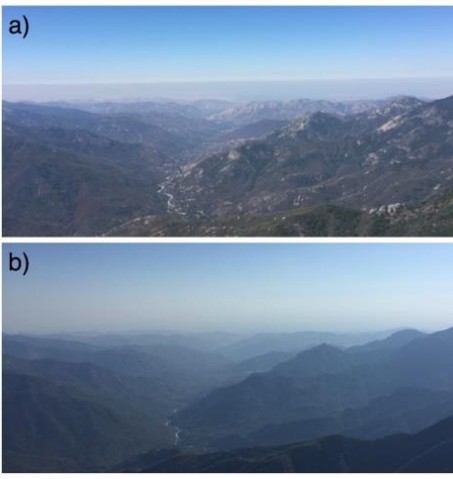

**Figure 3.** Looking toward the SJV from Moro Rock in SNP (36.5469 N, 118.7656 W; 2050 m ASL) at 11 am LT (panel a) and 5:30 pm LT (panel b). Photographs were taken by the authors on 29 June 2017.

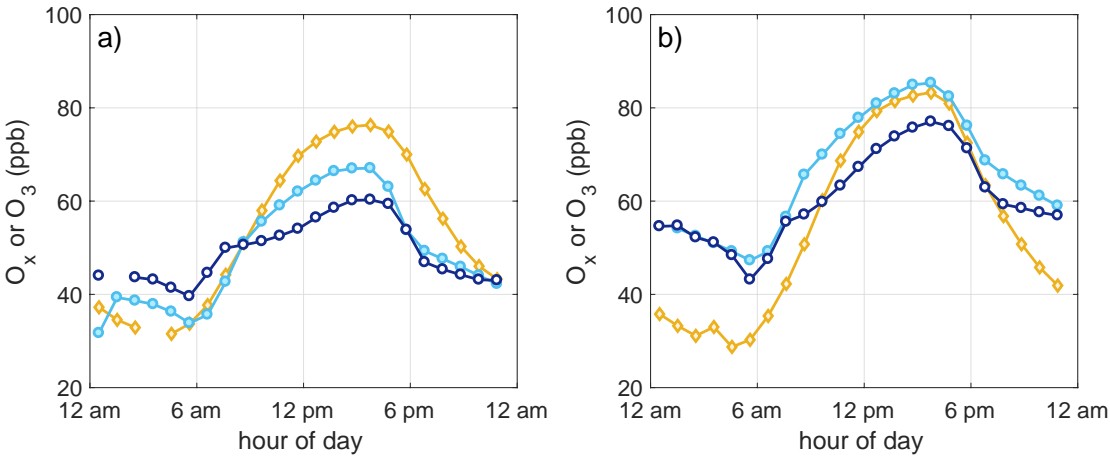

**Figure 4.** Hourly mean $O_x$ in Visalia (orange diamonds), SEQ1 (cyan filled circles), and SEQ2 (dark blue open circles) in springtime (panel a) and during $O_3$ season (panel b) 2001–2012. Data gaps are due to routine calibrations.

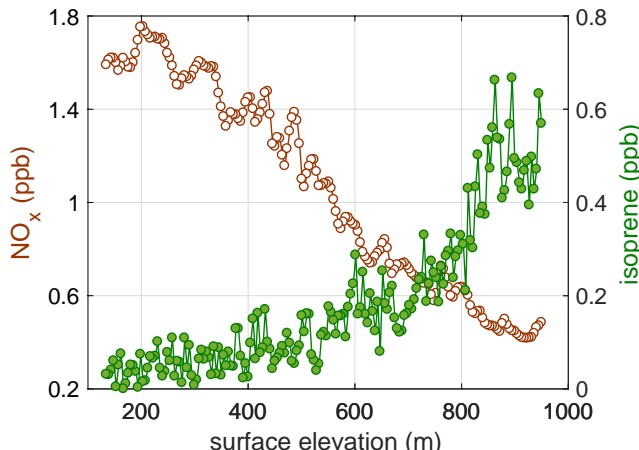

**Figure 5.** NOₓ (brown open circles) and isoprene (green filled circles) measured onboard the NASA DC-8 at ~10 am LT at the Sierra Nevada western slope from a mean altitude of 130 m AGL to 1000 m AGL on 19 June, 2016. The surface elevation is estimated by linearly interpolating across the total elevation change.

**Table 1.** Percent difference in afternoon (12–6 pm LT) $O_x$ or $O_3$ on weekdays and weekends calculated as: $(O_{x,weekday} - O_{x,weekend}) / O_{x,weekday}$ in Visalia, SEQ1, and SEQ2 in 2001–2003 on moderate and high temperature days. Errors are reported as standard errors of the mean.

| Temperature | Moderate | High |
|---|---|---|
| **$O_3$ season (June–October) 2001–2003** | | |
| | % | % |
| SEQ2 | 4.9 ± 3.9 | 3.5 ± 2.4 |
| SEQ1 | 4.6 ± 3.3 | 4.2 ± 1.9 |
| Visalia | −6.3 ± 3.5 | 5.3 ± 2.6 |
| **Springtime (April–May) 2001–2003** | | |
| | % | % |
| SEQ2 | 4.7 ± 5.5 | 5.2 ± 4.6 |
| SEQ1 | 3.5 ± 7.4 | 8.6 ± 4.9 |
| Visalia | −7.4 ± 4.6 | 12.2 ± 4.8 |

**Table 2.** $O_3$ changes in Visalia, SEQ1, and SEQ2 over 2001–2012 according to MD8A, SUM0, W126, and morning $O_x$ metrics based on a linear fit of annual mean data (shown in Figure 6) in the springtime and $O_3$ season. Each left column is the percent change with respect to fit value in 2001 at SEQ1 during $O_3$ season for comparison, which is the highest $O_3$ observed for each metric. Each right column is the fit slope

with slope errors in $O_3$ abundance units per year. Coloration is based on the TOAR categorization for trend significance (Lefohn et al., 2018), with p-values calculated using the Mann-Kendell non-parametric test: yellow, 0–0.05, statistically significant trend; green, 0.05–0.10, indicative of a trend; violet, 0.10–0.034, weak indication of change; and pink, 0.34–1, weak or no change.

| $O_3$ metric | MD8A | | SUM0 | | W126 | | Morning $O_x$ | |
|---|---|---|---|---|---|---|---|---|
| | **$O_3$ season (June–October)** | | | | | | | |
| | % | ppb yr$^{-1}$ | % | ppm h yr$^{-1}$ | % | ppm h yr$^{-1}$ | % | ppb yr$^{-1}$ |
| SEQ2 | −19 | −1.4 ± 0.41 | −15 | −1.2 ± 0.46 | −37 | −2.2 ± 0.72 | −17 | −1.0 ± 0.32 |
| SEQ1 | −13 | −1.0 ± 0.27 | −12 | −0.96 ± 0.21 | −28 | −1.7 ± 0.36 | −14 | −0.83 ± 0.21 |
| Visalia | −7 | −0.54 ± 0.30 | −3 | −0.20 ± 0.28 | −11 | −0.69 ± 0.41 | −6 | −0.50 ± 0.30 |
| | **Springtime (April–May)** | | | | | | | |
| | % | ppb yr$^{-1}$ | % | ppm h yr$^{-1}$ | % | ppm h yr$^{-1}$ | % | ppb yr$^{-1}$ |
| SEQ2 | −13 | −1.0 ± 0.38 | −16 | −1.2 ± 0.47 | −30 | −1.8 ± 0.62 | −13 | −0.78 ± 0.34 |
| SEQ1 | −8 | −0.59 ± 0.42 | −6 | −0.50 ± 0.53 | −24 | −1.5 ± 0.62 | −6 | −0.35 ± 0.32 |
| Visalia | −3 | −0.23 ± 0.39 | −4 | −0.31 ± 0.38 | −11 | −0.69 ± 0.49 | −8 | −0.39 ± 0.35 |

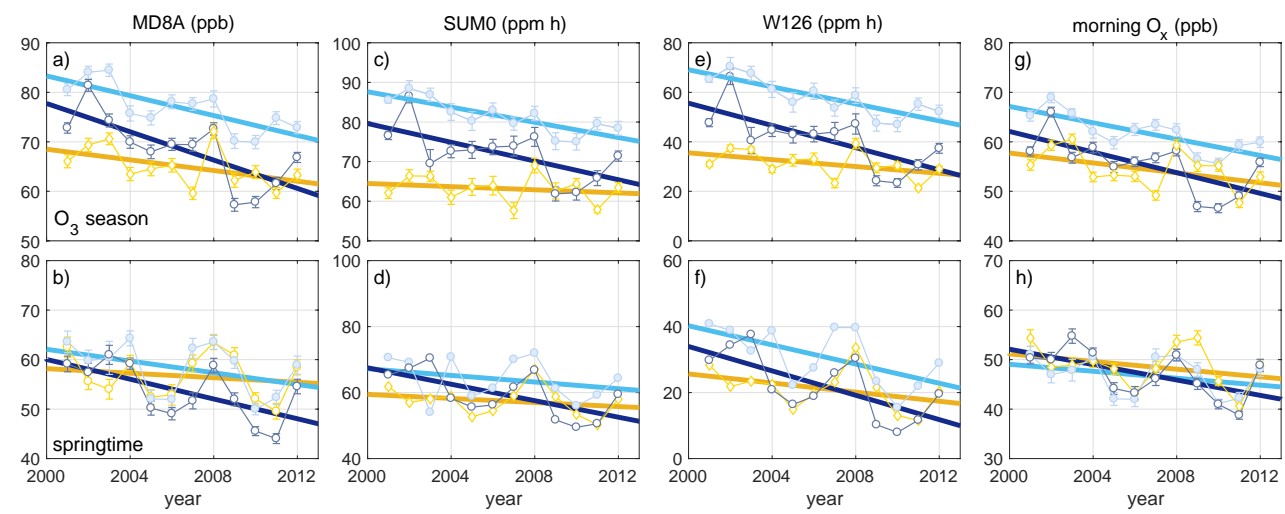

**Figure 6.** $O_3$ trends in Visalia (orange diamonds), SEQ1 (cyan filled circles), and SEQ2 (dark blue open circles) computed using MD8A (a–b), SUM0 (c–d), W126 (e–f), and morning $O_x$ (g–h) metrics during $O_3$ season (top row) and springtime (bottom row). Both MD8A and morning $O_x$ are computed as seasonal averages. Error bars in panels a–b and g–h are standard errors of the mean. Error bars in panels c and e are standards errors of the mean of the three $O_3$ season 3-month summations.