# Peer review of "On the effect of upwind emission controls on ozone in Sequoia National Park"

_Atmospheric Chemistry and Physics, 2017_

## Referee Comment (RC1) · Anonymous Referee #1 · 23 Jan 2018

This is a well-written manuscript, supported by ample and well-developed data analysis, that involves a topic of substantial interest. Thanks to the authors for preparing the work. A few minor suggestions and comments are provided simply for consideration.

General:

1) Am a little concerned about the conclusion regarding O3 sensitivity to downwind *distance* from Visalia/SJV given that (if i understood correctly) that this finding is based on just the two sites in SNP (one ∼ 10km downwind of the other). Two general concerns here:

a) Do the authors intend for readers to extrapolate this conclusion beyond these two specific SNP sites? Would a hypothetical 3rd site further downwind by 10-20 km from

[Figure]

SJV be expected to have shown even greater responsiveness? At some point down-wind, this conclusion presumably breaks down as areas become less and less influenced by SJV. If this conclusion is intended to be limited to these two SNP sites, maybe those parts of the paper that cite downwind distance as a factor in responsiveness could be revised to limit this conclusion to the SEQ1 and SEQ2 sites.

b) Is it possible, especially given complex flows in the region, that the elevation differences between these two sites is an equal or greater driver of the differential O3 decreases in the area than distance?

2) A little more detail on the approach used to project future exceedances would be useful (Table 5). Clearly, the trend is assumed constant, but then did you also assume that the within-year variability would also stay the same?

Specific:

Figure 2: Information about mean vector flow would be more informative than mean direction. The paper notes the mean wind speeds on page 6, line 3.

Section 2: If possible, a schematic of the various flows and layers would be valuable.

Page 5, line 2: "during *the* ozone season"

Page 6, line 1: "Figure 2"

Page 7, line 18: The 8-hour ozone NAAQS is slightly more involved than described here. EPA defines a design value metric to determine an area's status relative to the NAAQS. For 8-hour ozone, the design value is the 3-year average of the 4th-highest max daily 8-hour ozone concentration at a site. Based on the description here, it's not entirely clear what metric was used for the trends in table 1. Clarification would be helpful.

Page 7, lines 21-26: My (limited) understanding is that statistics like W126 are typically calculated over a specified period (e.g., 3 months for W126 as discussed on page 8).

Does this paper follow those conventions for the Table 1 trends? Either way, may want to clarify.

Page 8, lines 1 and 11: It's not immediately clear to me ... is the term "interannual variability" as used in the context of Table 1 referring to the year-to-year differences in these metrics (i.e., the standard deviation of yearly values over the 12-year period)? Or is it just referring to the trend itself as "interannual variability"? May want to clarify, especially if you mean the latter.

Page 9, line 21: The paper tends to reserve the use of the term "impacts" for O3 impacts on plants and use the term concentrations when talking about non-plant effects (e.g., human health). This is fine, but of course both are impacts.

Table 5: Given the statement on page 10, lines 1-3 about the potential overly optimistic nature of the W126 metric relative to SUM0, why not include SUM0 in Table 5 instead of W126?

Table 5: Very minor It might make it easier on the reader if this table was reconfigured such that directional changes were consistent across the two metrics (i.e., lower numbers indicate improvement). May want to consider switching from # of days required for an exceedance to something like the inverse of that.

Table 5: Per an earlier comment, it's not clear to me how you could have a value > 92 days (e.g., the value of 107 listed for 9 ppm h in 2021) if the W126 metric is calculated over 3 months. Wouldn't that be a "never"?

Page 11, lines 13-14: May want to clarify these specific listed values in text are for SUM0.

---

## Referee Comment (RC2) · Anonymous Referee #2 · 14 Apr 2018

General comments:

Overall, the paper is well written and is an easy read, but there are some fundamental issues that must be addressed before this paper can be published. In general, there are numerous, rather bold statements, that need to be substantiated. Most things are overstated in the manuscript, and the rudimentary analysis done in Section 3.4, past and future exceedances, is completely unacceptable for any paper that is going to be published in ACP, or any other scientific journal. As far as overstating goes, the first sentence of the abstract is simply not correct:

Abstract. Sequoia National Park (SNP) experiences the worst ozone (O3) pollution of any national park in the U.S.

My response to the first sentence of the abstract is: NO – if you look at the NPS ozone data for all of their sites, you find that Joshua Tree is actually the worst, Sequoia and Kings Canyon is comparable at best. Even though they are only using data through 2012, the following is still relative and the patterns remain the same: Acadia and Joshua Tree recently reported the highest ozone levels in 2017 for all NPS sites, and Yosemite also beat out Sequoia and Kings Canyon for 2017. Moreover, Dinosaur National Monument has wintertime ozone levels can greatly exceed what is observed at Sequoia and Kings Canyon. My point, the information disseminated throughout the manuscript must be conveyed accurately, and not overstated. Simply changing the sentence to "some of the worst" is all that is needed, but these types of statements are too common throughout the manuscript.

Additionally, the authors don't really convey any new information – one of their main points, that the transport of $NO_x$ is more important than the transport of ozone to the sites in the park is something people already know and understand for this area. How else would you manage to get higher levels in the park if the precursors were not being transported out of the source region photochecmically processed along the way?

Additionally, it is stated in a couple places in the manuscript that emission controls are optimized for the hottest days in the summer, so the policies that have been implemented are not optimized for decreasing springtime ozone, when it is cooler. This is a rather bold and cavalier statement to make without providing any type of information on what the policies are, and how the seasonal differences in temperature affect the emissions control strategies. In my opinion, there needs to be a substantial discussion, that pulls in what the policies are, and how the overall emissions are affected by these seasonal temperature differences to justify their statement that these polices are less effective in the springtime during cooler weather. My guess is they are trying to rehash the points about temperature dependence as as described in Pusede et al. (2015); however, they have stated that it's the emission control strategies that aren't optimized, so this means diving into the SIPs and seeing what and how emissions were/are con-

trolled and correlating this to the seasonal temperature changes. The authors beat on policy not being appropriate for the seasonal changes, so this needs to be addressed. In particular, what part of the SIPs are not effective for the springtime emissions and how can they demonstrate this? What would be done differently to improve the effectiveness of the emission control policies to improve springtime ozone?

Moreover, the authors discuss how precursor emission controls have been less effective at reducing O3 concentrations in SNP in springtime, yet, there is no mention or discussion about other factors that may be influencing springtime ozone. For example, how do the springtime chemistry and dynamic processes of the widely observed springtime maximum of ozone in the Northern Hemisphere mid latitudes influence ozone levels in this region? Are these processes influencing what the authors are referring to as less effective emission controls during the spring? Also, it's not actually clear in the paper how the authors get to the conclusion that "...precursor emission controls have been less effective at reducing O3 concentrations in SNP in springtime...".

Finally, the term "trend analysis" is used quite a bit in the paper; however, it would be useful if they included a figure of the full time series of ozone data from the sites, the annual 8-hr 4th high, a table of annual basic statistics to help set the stage for the analysis. What is presented is rather "thin" – the reader needs to be provided more information in order to better evaluate what is presented...which is very little. This is even that much more important for Sect. 3.4 – the authors should, at minimum, show the simple regression that was used to come up with the values in Table 2. I personally think this section should be removed or done in a much more rigorous manner, but the authors need to show how these values were derived.

Specific comments: P1, L21-22: If you are referring to the whole area (re Sierra Nevada forests), then you should use the 4 letter NPS designation for the site, SNP should be referred to as SEKI, as the measurements are representative of Sequoia and Kings Canyon NPs.

P1, L25-26: The reference cited in this sentence does not make the statement that it Sequoia is the most ozone polluted park in the U.S. – please ensure that you accurately represent what a reference says, period.

Sequoia National Park (SNP) is a unique and treasured ecosystem that is also the most ozone-polluted national park in the U.S. 25 (Meyer and Esperanza, 2016).

P2,L7-9: Revise the following sentence – reads awkwardly: On multi-decadal timescales, O3-resistant plants may thrive over O3-sensitive species, system-level dynamics that would maintain forest productivity and carbon storage, but would induce changes in ecosystem composition (Wang et al., 2016).

P3, L3: there are additional references that should be included regarding the W126 metric

P3,L16; technically, the NPS started measuring ozone in the early 1980s, not the late 1980s. Shenandoah NP started in 1983 and Sequoia and Kings Canyon NP – Lower Kaweah started in 1984.

P4, L25: Beginning a sentence with "Due to" is grammatically incorrect. The Chicago Manual of Style suggests using "due to" when you can replace it with "attributable to," but not when you could use "because of"; if a sentence starts with "due to", it is most likely incorrect. Therefore, please revise.

P4, L30-31: "strongly" in "strongly temperature-dependent" really should be defined here – a counter to this statement is that in the Uintah Basin, during snow cover cold periods, ozone levels are usually higher there than in the Sequoia and Kings Canyon National Parks, yet the temperature is significantly lower. Temperature is only one factor, not the only factor. Moreover, high ozone episodes have occurred as early as March, but high ozone starting in the spring is fairly typical, so I would change "summer" to "spring" or through the fall. Ozone levels in Sequoia and Kings Canyon are comparable in April and September.

P5, L4: "due to" is inappropriate here – it is a result of the Mediterranean climate.

P5, L21: First off, there isn't a methods or experimental section – more information needs to be provided. This get to the more important point that you state that all data are provided by CARB, when in fact, they are not. The data are served up by CARB on their website, but the NPS data is provided by the NPS on their data page, which also gets uploaded to AQS, which is the main repository that houses all national ozone data – it is the single main repository. CARB either serves up the data directly from AQS or in their own database, where they have either obtained the data directly from the NPS or AQS. So, it should be clearly stated who is the proprietor of the data and where it was obtained from. These have been merged into one item, and what is disseminated in this sentence is incorrect.

P7, L15: The NAAQS for ozone is an 8 hour average value; it is the annual 4th highest daily maximum 8 hour average ozone concentration, averaged over 3 consecutive years (the design value) – this must not exceed 70 ppb. So, what you are saying is repetitive and not conveyed properly. What you should say is the annual 4th highest daily maximum 8 hour average, or the DM8HA or DM8A, not the "8-h O3 NAAQS".

P7, L16: Why are trends only reported as a percent change? It would be more useful to include the ppb per yr trend in the table, along with the percent change. Moreover, why is only the 8-hr daily max being listed? I'm assuming it's the annual 4th high daily max 8-hr average, but it's not stated in the text. Please clarify.

P7, L16: This sentence is not correct – see previous comment. "The 8-h O3 NAAQS is a human health-based metric computed as the maximum unweighted daily 8-h average O3 mixing ratio." As for SUM0 – you need to say why it's called SUM0 – as in, why is it a "0"?

P8, L1-11: For this paragraph, only percentages are reported – it is absolutely necessary to include what the corresponding values were on ppb and ppm hrs for W126 and SUM0. For this to have value to ecosystem and plant effects folks, numbers, not

percentages, are needed.

Section 3.4 You state that you "predict future O3 levels in the context of protective threshold"; however, it is not stated how your do this in this section – please provide necessary information and figures.

"Future exceedances are computed assuming individual daily indices continue to decline at the 2001–2012 rate and are projected from 2011–2012 values." Is this a reasonable assumption? I'm not convinced this is the case. There is ozone data beyond 2012 at these sites, so does the rate of decline hold true? The fact that you are predicting future ozone levels off of this would suggest it should be evaluated.

"If past decreases in O3 continue over the next two decades, we predict no exceedances of the 8-h O3 NAAQS at SEQ2 by 2021 in springtime and by 2031 during O3 season, no exceedance of the 9-ppm h W126 threshold by 2021, and no further exceedances of 5- and 7-ppm h thresholds by 2031."

Following suit, more information needs to be provided to make such a bold statement when using such a rudimentary method. How much is NOx going to go down? How are large scale circulation patterns (e.g., PDO, ENSO, etc.) going to change and influence what is being transported in? What about the different climate futures? There are an array of emissions scenarios that can lead to significant differences in what you are inaccurately and inappropriately conveying here. Also, climate change - This section either must be expanded upon significantly or simply removed from the paper. As an example that contradicts your statements about ozone exceedances, the following is pulled directly from Val Martin et al. (2015) for Sequoia and Kings Canyon. Here, the report the actual values using different climate futures in order to assess what the ozone and W126 values will be. According to their rigorous analysis, both ozone and W126 values will exceed the current NAAQS level of 70 ppb and W126 values will also increase, and be well above the 5-9 ppm hr range.

Summer MDA-8 Ozone (ppb)

2000 (Baseline): 71.3, 2050 (RCP4.5): 72.9, 2050 (RCP8.5): 73.8

O3 W126 (ppm-hr)

2000 (Baseline): 46.0, 2050 (RCP4.5): 50.6, 2050 (RCP8.5): 53.2

Val Martin, M., C. L. Heald, J.-F. Lamarque, S. Tilmes, L. K. Emmons, and B. A. Schichtel How emissions, climate, and land use change will impact mid-century air quality over the United States: a focus on effects at national parks, Atmos. Chem. Phys., 15, 2805–2823, 2015, www.atmos-chem-phys.net/15/2805/2015/.

P9, L30: "O3 reductions predicted by W126 are almost twice those of SUM0." What does this statement mean? How are ozone reductions predicted by W126 or SUM0? Both of these metrics are determined from ozone levels – how are these used to predict ozone reductions?

P10,L2: Regarding the following statement: "...W126 likely provides an overly optimistic representation of past and future trends in O3 impacts in SNP.", this is a rather bold statement to make to summarize the paragraph, yet you provide no hard evidence of this – there is nothing in this section that supports this statement. Please address this in a more rigorous manner.

P10, L9: For the following statement: "...leading to policies not optimized to decrease O3 in cooler springtime conditions."

Please elaborate on this point - this needs to be shown quantitatively. How large or small of a difference are you suggesting? What are the policies? How are they not optimized for the cooler springtime conditions? What could/should be done to address this policies in order to optimize them for the springtime?

P10,L15: Regarding the following "Third, aircraft observations collected in the direction of daytime upslope flow from the SJV to Sierra Nevada foothills reveal substantial decreases in NOx concentrations relative to isoprene, a key contributor to total organic reactivity (e.g., Beaver et al., 2012)." You are consding the 2001-2012 time frame, how

representative is this single day? Can this be put in to greater context?

P10, L18: For the following sentence: "This implies that PO3 in Visalia and SNP is differently sensitive to emission controls, with SNP more responsive to NOx emissions control than Visalia."

This is only one aspect of the issue, the other is that you are sitting in a source region, so the regime you are in is different; also, there is mixing and dilution that occur with transport, so this is another major factor - it's not simply response to emissions controls. This needs to be addressed and put into context.

P11, L15-16: "...day due the mixing..." please fix this sentence, and it would be best not to use due to...

As it currently stands, the data disseminated in the tables is not very useful, especially Table 2. What would be better to provide in Table 2 are the projected DM8HA values in ppb and the W126 values in ppm hrs, along with their corresponding #s of exceedances per year. However, the method used for this work is not suitable for providing any type of reasonable predicted value. As for Table 1, actual values should be included along with the percent change.

In summary, before this paper is worthy of being published, there are significant issues that must be addressed.

---

## Author Comment (AC1) · 17 Jul 2018

**Author response**
On the effect of upwind emission controls on ozone in Sequoia National Park

**Referee Comments (bold)**

5 *Author Response (italics)*

Author Changes to Manuscript (standard)

We thank the reviewers for their feedback, which has improved the quality of the manuscript. We address each point below.

10 **Anonymous Referee #1**

**This is a well-written manuscript, supported by ample and well-developed data analysis, that involves a topic of substantial interest. Thanks to the authors for preparing the work. A few minor suggestions and comments are provided simply for consideration.**

**General:**

15 **1) Am a little concerned about the conclusion regarding O3 sensitivity to downwind \*distance\* from Visalia/SJV given that (if i understood correctly) that this finding is based on just the two sites in SNP (one ~ 10km downwind of the other). Two general concerns here:**

**a) Do the authors intend for readers to extrapolate this conclusion beyond these two**
20 **specific SNP sites? Would a hypothetical 3rd site further downwind by 10-20 km from SJV be expected to have shown even greater responsiveness? At some point downwind, this conclusion presumably breaks down as areas become less and less influenced by SJV. If this conclusion is intended to be limited to these two SNP sites, maybe those parts of the paper that cite downwind distance as a factor in responsiveness could be**
25 **revised to limit this conclusion to the SEQ1 and SEQ2 sites.**

*We have adjusted the way we speak about downwind distance and limited our conclusions to SEQ1 and SEQ2. We also add text to address this comment directly in the text.*

Page 3, Lines 31–32: "We describe these $O_3$ changes in Visalia and SNP as function of distance downwind of Visalia by way of data collected at two monitoring stations located on the western slope of the Sierra
30 Nevada Mountains."

Page 8, Lines 32–33: "…$O_3$ decreased more rapidly in SNP versus Visalia and at SEQ2 versus SEQ1."

Page 9, Line 12: "Additionally, greater $O_3$ decreases were observed at SEQ1 than Visalia and at SEQ2 compared to SEQ1."

Page 10, Lines 2–3: "$NO_x$ decreases have generally made greater improvements in $O_3$ in SEQ1 than Visalia and in SEQ2 than SEQ1, a trend that corresponds to increasing distance downwind of the SJV."

Page 11, Lines 14–20: "Downwind sites usually experience $PO_3$ chemistry that is more $NO_x$-limited than in the often $NO_x$-suppressed (or at least more $NO_x$-suppressed) urban core. As a result, we expect similar location-specific $O_3$ trends in other ecosystems and national parks downwind of major $NO_x$ sources like cities. However, while the extent of observed $O_3$ improvements in SNP follows the pattern of increasing distance downwind of Visalia with sustained $NO_x$ emission control in the SJV (Russell et al., 2010; Pusede and Cohen, 2012), $PO_3$ chemistry is non-linear and the direction of location-specific trends may vary. That said, at some distance downwind this conclusion breaks down, as areas become less and less influenced by the upwind source."

**b) Is it possible, especially given complex flows in the region, that the elevation differences between these two sites is an equal or greater driver of the differential O3 decreases in the area than distance?**

*You are correct. The distance of airflow will be dictated by the mountain terrain and will travel a distance longer than determined as a straight-line path on a flat surface. Accounting for the change in elevation simply using the Pythagorean theorem, the horizontal change is greater than the vertical change to the extent that the horizontal distance is a reasonable approximation. We have added text to this point.*

Page 6, Lines 9–16: "If $O_3$ attributed to local $PO_3$ in Visalia is greatest around 2 pm LT, typical of many urban locations, with mean winds at SEQ1 of 3 m s$^{-1}$ and SEQ2 of 2 m s$^{-1}$, we expect $O_3$ to peak in SEQ1 at ~5 pm (45 km downwind of Visalia) and at SEQ2 shortly after (9.7 km downwind of SEQ1, which includes the change in elevation using the Pythagorean theorem). This is broadly what we observe. While the actual distance of airflow is dictated by the mountain terrain and a parcel of air will travel a distance longer than the straight-line path on a smooth surface, the timing of the $O_3$ diurnal patterns is consistent with airflow travel time roughly equal to that determined by the horizontal distance and mean wind speed. There has been no change in the hour of peak $O_3$ mixing ratio at either SEQ1 or SEQ2 over the 2001 to 2012 period."

**2) A little more detail on the approach used to project future exceedances would be useful (Table 5 [sic]). Clearly, the trend is assumed constant, but then did you also assume that the within-year variability would also stay the same?**

*We have removed future projections from our analysis in response to comments from anonymous Reviewer 2.*

**Specific:**

**3) Figure 2: Information about mean vector flow would be more informative than mean direction. The paper notes the mean wind speeds on page 6, line 3.**

*We have added wind rose plots over SEQ1 and SEQ2 with mean wind speeds shown. In the process, we also discovered an error, in which SEQ1 and SEQ2 were mislabeled. New Figure 2:*

[Figure]

[Figure]

"**Figure 2.** Hourly mean wind directions in Visalia (orange diamonds), SEQ1 (cyan filled circles), and SEQ2 (dark blue open circles) in April–October, 2001–2012 (panel a). Wind rose for SEQ1 (panel b) with the direction of the neighboring sites of Visalia (orange), SEQ1 (cyan), and SEQ2 (dark blue) indicated."

**4) Section 2: If possible, a schematic of the various flows and layers would be valuable.**

*Because we do not advance knowledge of airflow in the SJV, we prefer not include a schematic. We direct the reviewer to the excellent diagram in Zhong et al. (2004).*

Page 4, Line 16: "Multiple airflow patterns influence $O_3$ in SNP and the SJV (see Zhong et al. (2004) for a diagram)."

**Page 5, line 2: "during \*the\* ozone season"**

*Corrected.*

5  **Page 6, line 1: "Figure 2"**

*Corrected.*

**Page 7, line 18: The 8-hour ozone NAAQS is slightly more involved than described here. EPA defines a design value metric to determine an area's status relative to the NAAQS. For 8-hour ozone, the design value is the 3-year average of the 4th-highest max daily 8-**
10  **hour ozone concentration at a site. Based on the description here, it's not entirely clear what metric was used for the trends in table 1. Clarification would be helpful.**

*We have added a definition of non-attainment, changed NAAQS to MD8A $O_3$ in numerous places throughout the text, and defined our use of exceedance.*

Page 7, Lines 31–34; Page 8, Line 1: "The MD8A $O_3$ is a human health-based metric computed as the
15  maximum unweighted daily 8-h average $O_3$ mixing ratio. A region is classified as in nonattainment of the NAAQS when the fourth-highest MD8A $O_3$ over a 3-yr period, known as the design value, exceeds a given standard. In this work, we utilize the seasonal mean MD8A and discuss $O_3$ exceedances as individual days in which MD8A $O_3 > 70.4$ ppb, the current 8-h NAAQS."

**8) Page 7, lines 21-26: My (limited) understanding is that statistics like W126 are**
20  **typically calculated over a specified period (e.g., 3 months for W126 as discussed on page 8). Does this paper follow those conventions for the Table 1 trends? Either way, may want to clarify.**

*We have changed our previous computations of daily SUM0 and W126 indices to consecutive 3-month summations, following EPA protocol for W126. We have clarified*
25  *this in the text.*

Page 8, Lines 8–18: "Here, SUM0 and W126 summations are computed following the W126 protocol (Environmental Protection Agency, 2016), affording straightforward comparisons between the metrics. First, in months with less than 75% of hourly data coverage in the 8 am–8 pm LT window, missing values are replaced with the lowest observed hourly measurement over the study period (i.e. April–October) only until
30  the dataset is 75% complete. Second, monthly summations of daily indices, comprised of hourly data (8 am– 7 pm), are computed; when data are missing, the summation is divided by the data completeness fraction.

Consecutive 3-month metrics are computed by adding monthly indices. In practice, SUM0 and W126 are computed as 3-yr averages of the highest 3-month summation; however, we define springtime SUM0 and W126 as the 3-month summation over April–June and $O_3$ season SUM0 and W126 as the mean of the 3-month summations over June–August, July–September, and August–October (not the highest of the three 3-month sums). Because less than 15% of data were available for August 2008 at SEQ1, $O_3$ season SUM0 and W126 were computed as the mean of 3-month summations over June, July, and September, and July, September, and October only for this site and year."

**9) Page 8, lines 1 and 11: It's not immediately clear to me ... is the term "interannual variability" as used in the context of Table 1 referring to the year-to-year differences in these metrics (i.e., the standard deviation of yearly values over the 12-year period)? Or is it just referring to the trend itself as "interannual variability"? May want to clarify, especially if you mean the latter.**

*We have exchanged use of "variability" for "trends" to make this distinction clear and renamed the subsection.*

**10) Page 9, line 21: The paper tends to reserve the use of the term "impacts" for $O_3$ impacts on plants and use the term concentrations when talking about non-plant effects (e.g., human health). This is fine, but of course both are impacts.**

*We added clarification to the initial usage but retained the convention for clarity.*

Page 5, Lines 18–20: "In this manuscript, for clarity we generally use the term *impacts* when discussing ecosystem metrics and *concentrations* when talking about human health metrics; $O_3$ ecosystem and human health effects are of course both $O_3$ impacts."

**11) Table 5: Given the statement on page 10, lines 1-3 about the potential overly optimistic nature of the W126 metric relative to SUM0, why not include SUM0 in Table 5 instead of W126?**

*There are no protective thresholds for SUM0, only for W126. We have removed the values for $O_3$ season, which we state are the poorest predictor of plant $O_3$ uptake. We have clarified as follows:*

Page 9, Line 26: "While there is no standard for SUM0, there are three time-integrated W126 protective thresholds."

**12) Table 5: Very minor It might make it easier on the reader if this table was reconfigured such that directional changes were consistent across the two metrics (i.e.,**

**lower numbers indicate improvement). May want to consider switching from # of days required for an exceedance to something like the inverse of that.**

*We have removed the table, placing the values in the text. We were unable to think of a clear way to present the inverse of days until exceedance. However, now that the data and explanation are in paragraph form, we hope the distinction is improved. New text:*

Page 9, Lines 29–32: "Rather than calculate W126 exceedances using a 3-month summation of monthly indices, we instead count the number of days required for an exceedance to occur, summing daily W126 indices from the first day of the springtime (1 April). A larger number of days indicates improved air quality. We do this to generate information in addition to exceedance frequency, as W126 $O_3$ at SEQ1 and SEQ2 is greater than all three standards in all years in both seasons."

**13) Table 5: Per an earlier comment, it's not clear to me how you could have a value > 92 days (e.g., the value of 107 listed for 9 ppm h in 2021) if the W126 metric is calculated over 3 months. Wouldn't that be a "never"?**

*This has been removed.*

**14) Page 11, lines 13-14: May want to clarify these specific listed values in text are for SUM0.**

*Reference to ~18% reduction in $O_3$ season at SEQ2 encompasses the reductions across all three metrics of 19% (MD8A), 18% (SUM0), and 17% (Morning $O_x$). We have changed the text to make this clear:*

Page 13, Lines 27–29: "The three metrics, MD8A, SUM0, and morning $O_x$, all indicate comparable reductions in $O_3$ over 2001–2012, with decreases of ~7% (springtime) and ~13% ($O_3$ season) at SEQ1 and 13–16% (springtime) and 15–19% ($O_3$ season) at SEQ2."

**Anonymous Referee #2**

**General comments:**

**Overall, the paper is well written and is an easy read, but there are some fundamental issues that must be addressed before this paper can be published. In general, there are numerous, rather bold statements, that need to be substantiated. Most things are overstated in the manuscript, and the rudimentary analysis done in Section 3.4, past and future exceedances, is completely unacceptable for any paper that is going to be published in ACP, or any other scientific journal.**

**1) As far as overstating goes, the first sentence of the abstract is simply not correct:**

**"Abstract. Sequoia National Park (SNP) experiences the worst ozone ($O_3$) pollution of any national park in the U.S." [quotations added]**

**My response to the first sentence of the abstract is: NO – if you look at the NPS ozone**
5 **data for all of their sites, you find that Joshua Tree is actually the worst, Sequoia and Kings Canyon is comparable at best. Even though they are only using data through 2012, the following is still relative and the patterns remain the same: Acadia and Joshua Tree recently reported the highest ozone levels in 2017 for all NPS sites, and Yosemite also beat out Sequoia and Kings Canyon for 2017. Moreover, Dinosaur National**
10 **Monument has wintertime ozone levels can greatly exceed what is observed at Sequoia and Kings Canyon. My point, the information disseminated throughout the manuscript must be conveyed accurately, and not overstated. Simply changing the sentence to "some of the worst" is all that is needed, but these types of statements are too common throughout the manuscript.**

15 *Changed to:*

Abstract, Line 1: "Ozone ($O_3$) pollution in Sequoia National Park (SNP) is among the worst of any national park in the U.S."

Page 1, Lines 25–26: "Sequoia National Park (SNP) is a unique and treasured ecosystem that is also one of the most ozone-polluted national parks in the U.S. (National Park Service, 2015a)."

20 National Park Service, 2009–2013 Ozone estimates for parks, http://www.nature.nps.gov/air/Maps/AirAtlas/IM_materials.cfm, last access 7 July 2018, 2015a.

**2) Additionally, the authors don't really convey any new information – one of their main points, that the transport of NOx is more important than the transport of ozone to the sites in the park is something people already know and understand for this area.**
25 **How else would you manage to get higher levels in the park if the precursors were not being transported out of the source region photochecmically processed along the way?**

*We agree past analyses have identified precursor transport as important and stated that fact in the initial submission with reference to Jacobson (2001). Our focus is on observed trends in $O_3$ over time and differences in those trends with season, which to our knowledge*
30 *have not yet been published. Our main point is not that $NO_x$ transport is more important than $O_3$ transport, but that because of this, $O_3$ chemistry in the SJV and SNP, and hence $O_3$ concentrations, are differently sensitive to $NO_x$ emission control.*

*In addition, the influence of $NO_x$ transport on SNP $O_3$ has not yet been shown empirically to our knowledge.*

**3) Additionally, it is stated in a couple places in the manuscript that emission controls are optimized for the hottest days in the summer, so the policies that have been implemented are not optimized for decreasing springtime ozone, when it is cooler. This is a rather bold and cavalier statement to make without providing any type of information on what the policies are, and how the seasonal differences in temperature affect the emissions control strategies. In my opinion, there needs to be a substantial discussion, that pulls in what the policies are, and how the overall emissions are affected by these seasonal temperature differences to justify their statement that these polices are less effective in the springtime during cooler weather. My guess is they are trying to rehash the points about temperature dependence as described in Pusede et al. (2015); however, they have stated that it's the emission control strategies that aren't optimized, so this means diving into the SIPs and seeing what and how emissions were/are controlled and correlating this to the seasonal temperature changes. The authors beat on policy not being appropriate for the seasonal changes, so this needs to be addressed. In particular, what part of the SIPs are not effective for the springtime emissions and how can they demonstrate this? What would be done differently to improve the effectiveness of the emission control policies to improve springtime ozone?**

*We state that controls are designed to address high $O_3$ as defined by the 8-h NAAQS. We also state that in the SJV, these exceedances are most frequent when temperatures are hottest. We do not say that controls are just optimized for hot days.*

*Pusede et al. (2015) is a review paper on the body of literature describing the $O_3$-temperature correlation, we are not sure in what respect such a paper can be rehashed.*

*We have added text and references to EPA guidance for modeling to be used for regulatory design to select $O_3$ episodes in which the MD8A is high. We have also elaborated on episode selection as relevant.*

Page 12, Lines 3–24: "Over 2001–2012, $O_3$ declines have mostly been smaller in SNP when plant $O_3$ uptake is greatest (springtime), despite comparable $NO_x$ decreases in both seasons. This may be in part because regulatory strategies prioritize attainment of the $O_3$ NAAQS in polluted urban areas like the SJV basin, where air parcels influenced by the results of these controls are then transported downwind to locations with different $PO_3$ chemistry. In the development of regulatory plans, agencies use models to hindcast past $O_3$ episodes, facilitating testing of the efficacy of specific $NO_x$ and/or organic emissions reductions over that episode to meet the 8-h $O_3$ NAAQS or progress goals (Environmental Protection Agency, 2007; Environmental Protection Agency, 2014). In nonattainment areas, U.S. EPA guidance recommends modeling past time

periods that meet a number of specific criteria, such as typifying the meteorological conditions that correspond to high $O_3$ days as defined by the MD8A greater than the NAAQS value and focusing on the ten highest modeled $O_3$ days (Environmental Protection Agency, 2007; Environmental Protection Agency, 2014).

5 Regulatory modeling in the SJV (Visalia, SEQ1, and SEQ2 are included in this attainment demonstration) is more comprehensive, as it was recently updated to span the full $O_3$ season (defined as May–September); still potential reductions (known as relative reduction factors, RRFs) are based on the MD8A and restricted to high $O_3$ days (San Joaquin Valley Air Pollution Control District, 2007; San Joaquin Valley Air Pollution Control District, 2014). In the SJV, high $O_3$ days are most frequent in the late summer ($O_3$ season) and on the hottest days of the year (Pusede and Cohen, 2012). Even in SEQ1 and SEQ2, days with MD8A > 70.4 ppb

10 are far more common in the summer. Because of chemical and meteorological differences between seasons, this may lead to policies not optimized to decrease $O_3$ in cooler springtime conditions, which in the SJV are more $NO_x$-suppressed and therefore more sensitive to controls on reactive organic compounds (Pusede et al., 2014). In addition, we observe greater year-to-year $O_3$ variability in the springtime than during $O_3$ season (Figure 6), suggestive of a larger relative role of interannual meteorological variability controlling $O_3$. Deeper

15 cuts in emissions would be required in the springtime, as decreases in anthropogenic emissions have a proportionally smaller effect on the total $O_3$ abundance than during $O_3$ season."

Environmental Protection Agency: Guidance on the use of models and other analyses for demonstrating attainment of air quality goals for ozone, $PM_{2.5}$, and regional haze, EPA-454/B-07-002, Research Triangle Park, NC, 2007.

20 Environmental Protection Agency: Draft modeling guidance for demonstrating attainment of air quality goals for ozone, $PM_{2.5}$, and regional haze, Research Triangle Park, NC, 2014.

San Joaquin Valley Air Pollution Control District, 2016 Plan for 2008 8-hour ozone standard: http://www.valleyair.org/Air_Quality_Plans/Ozone-Plan-2016.htm, 2016.

San Joaquin Valley Unified Air Pollution Control District, 2007 Ozone plan:
25 http://www.valleyair.org/Air_Quality_Plans/AQ_Final_Adopted_Ozone2007.htm, 2007.

**4) Moreover, the authors discuss how precursor emission controls have been less effective at reducing $O_3$ concentrations in SNP in springtime, yet, there is no mention or discussion about other factors that may be influencing springtime ozone. For example, how do the springtime chemistry and dynamic processes of the widely observed springtime**
30 **maximum of ozone in the Northern Hemisphere mid latitudes influence ozone levels in**

**this region? Are these processes influencing what the authors are referring to as less effective emission controls during the spring? Also, it's not actually clear in the paper how the authors get to the conclusion that ". . .precursor emission controls have been less effective at reducing $O_3$ concentrations in SNP in springtime. . .".**

*The initial submission included discussion of trans-Pacific transport and its influence on springtime $O_3$ trends. We have expanded the discussion to address trends in springtime background $O_3$ more broadly and included this text:*

Page 12, Lines 24–34; Page 13, Lines 1–14: "An additional challenge to regulators is the contribution of background $O_3$ concentrations to $O_3$ levels (Cooper et al., 2015), as natural sources produce $O_3$ even in the absence of anthropogenic precursor emissions, $O_3$ can be transported over significant distances, and $O_3$ concentrations are influenced by large-scale meteorological and climatic events. Multiple studies have identified an increasing trend in $O_3$ at rural sites (often used as a proxy for background $O_3$) in the western U.S., particularly in the springtime (e.g., Cooper et al., 2012, Lin et al., 2017). Parrish et al. (2017) presented observational evidence of a slowdown and reversal of this trend on the California west coast since 2000, though the reversal was stronger in the summer than springtime. Using observations and the GFDL-AM3 model, Lin et al. (2017) computed that Asian anthropogenic emissions accounted for 50% of simulated springtime $O_3$ increases at western U.S. rural sites, followed by rising global methane (13%) and variability in biomass burning (6%). Northern mid-latitude transport of Asian pollution to the western U.S. is strongest during March–April and weakest in the summertime (e.g., Wild and Akimoto, 2001; Liu et al., 2003; Liu et al., 2005), with high-elevation locations in the Sierra Nevada Mountains being more vulnerable to reception of Asian $O_3$ and $O_3$ precursors (e.g., Vicars and Sickman, 2001; Heald et al., 2003; Hudman et al., 2004). Hudman et al. (2004) compared surface observations with GEOS-Chem-modeled $O_3$ enhancements in Asian pollution outflow, finding that, on average, transport events in April–May 2002 led to $8 \pm 2$ ppb higher MD8A $O_3$ concentrations at SEQ2. East Asian $NO_x$ emissions have risen over our study window (e.g., Miyazaki et al., 2017), potentially causing an increase in the influence of trans-Pacific transport on $O_3$ concentrations at SEQ2 and reducing the efficacy of local $NO_x$ control in springtime. Background $O_3$ concentrations are also responsive to large-scale climatic events, and elevated springtime $O_3$ at rural sites in the western U.S. has been linked to strong La Niña winters (Lin et al., 2015; Xu et al., 2017), which are associated with an increased frequency of deep tropopause folds that entrain $O_3$-rich stratospheric air into the troposphere (Lin et al., 2015). Over our study period, strong La Niña events occurred during the winter of 2007–2008 and 2010–2011. In general, transport of Asian pollution and tropopause folds are expected to have a greater impact in the springtime and at the higher-elevation SEQ2. While we do observe smaller decreases in $O_3$ in springtime at

SEQ2 than during O₃ season, interannual trends have been more downward at SEQ2 than at the lower elevation sites, SEQ1 and Visalia, in both seasons. This suggests that these factors may impact surface O₃ at high-elevations in SNP during individual events (e.g., Hudman et al., 2004) but that interannual trends in seasonal averages are more influenced by chemistry during upslope outflow from the SJV."

5  **5) Finally, the term "trend analysis" is used quite a bit in the paper; however, it would be useful if they included a figure of the full time series of ozone data from the sites, the annual 8-hr 4th high, a table of annual basic statistics to help set the stage for the analysis. What is presented is rather "thin" – the reader needs to be provided more information in order to better evaluate what is presented. . .which is very little. This is**
10 **even that much more important for Sect. 3.4 – the authors should, at minimum, show the simple regression that was used to come up with the values in Table 2. I personally think this section should be removed or done in a much more rigorous manner, but the authors need to show how these values were derived.**

*We did not find any use of the term "trend analysis" in the paper.*

15 *We have removed Section 3.4.*

*We have added a figure with the regressions used to produce Table 1 that also displays the basic statistics we discuss. We do not include the design value because that is not a focus of our study, a point that should be clear in the updated version.*

[Figure]

20 "**Figure 6.** O₃ trends in Visalia (orange diamonds), SEQ1 (cyan filled circles), and SEQ2 (dark blue open circles) computed using MD8A (a–b), SUM0 (c–d), W126 (e–f), and morning O$_x$ (g–h) metrics during O₃ season (top row) and springtime

(bottom row). Error bars in panels a–b and g–h are standard errors of the mean. Error bars in panels c and e are standards errors of the mean of the three $O_3$ season 3-month summations."

**Specific comments:**

**6) P1, L21-22: If you are referring to the whole area (re Sierra Nevada forests), then you should use the 4 letter NPS designation for the site, SNP should be referred to as SEKI, as the measurements are representative of Sequoia and Kings Canyon NPs.**

*We use data from two monitoring stations in SNP and do not use data from Kings Canyon. We do not know if these measurements are representative of the full SEKI and believe SNP is a more accurate descriptor for our purpose.*

**7) P1, L25-26: The reference cited in this sentence does not make the statement that it Sequoia is the most ozone polluted park in the U.S. – please ensure that you accurately represent what a reference says, period. "Sequoia National Park (SNP) is a unique and treasured ecosystem that is also the most ozone-polluted national park in the U.S. (Meyer and Esperanza, 2016)."**

*We have updated the reference:*

Page 1, Lines 25–26: "Sequoia National Park (SNP) is a unique and treasured ecosystem that is also one of the most ozone-polluted national parks in the U.S. (National Park Service, 2015a)."

National Park Service, 2009–2013 Ozone estimates for parks, http://www.nature.nps.gov/air/Maps/AirAtlas/IM_materials.cfm, last access 7 July 2018, 2015a.

**8) P2,L7-9: Revise the following sentence – reads awkwardly: On multi-decadal timescales, $O_3$-resistant plants may thrive over $O_3$-sensitive species, system-level dynamics that would maintain forest productivity and carbon storage, but would induce changes in ecosystem composition (Wang et al., 2016).**

*We have revised the sentence to read:*

Page 2, Lines 7–9: "On multi-decadal timescales, $O_3$-resistant plants may thrive over $O_3$-sensitive species, and these system-level dynamics would maintain forest productivity and carbon storage but would induce changes in ecosystem composition (Wang et al., 2016)."

**9) P3, L3: there are additional references that should be included regarding the W126 Metric.**

*We have included two additional EPA references:*

Page 3, Lines 5–6: "W126 is a 12-h daily 3-month summation weighted to emphasize higher $O_3$ concentrations (Environmental Protection Agency, 2006; Environmental Protection Agency, 2016) that is used by the U.S. National Park Service."

Environmental Protection Agency, Air quality criteria for ozone and related photochemical oxidants, Final report EPA/600/R-05/004aF-cF, Washington, DC, 2006.

Environmental Protection Agency, Ozone W126 index: https://www.epa.gov/air-quality-analysis/ozone-w126-index, last access: 27 October 2016, 2016.

**10) P3,L16; technically, the NPS started measuring ozone in the early 1980s, not the late 1980s. Shenandoah NP started in 1983 and Sequoia and Kings Canyon NP – Lower Kaweah started in 1984.**

*We removed the word "late" from this sentence.*

**11) P4, L25: Beginning a sentence with "Due to" is grammatically incorrect. The Chicago Manual of Style suggests using "due to" when you can replace it with "attributable to," but not when you could use "because of"; if a sentence starts with "due to", it is most likely incorrect. Therefore, please revise.**

*We have revised the sentence to read:*

Page 4, Lines 29–32: "The prevalence of shallow nighttime surface inversions in the SJV means that evening downslope valley flow at higher elevations may be stored within nocturnal residual layers and entrained into the surface layer the following morning."

**12) P4, L30-31: "strongly" in "strongly temperature-dependent" really should be defined here – a counter to this statement is that in the Uintah Basin, during snow cover cold periods, ozone levels are usually higher there than in the Sequoia and Kings Canyon National Parks, yet the temperature is significantly lower. Temperature is only one factor, not the only factor. Moreover, high ozone episodes have occurred as early as March, but high ozone starting in the spring is fairly typical, so I would change "summer" to "spring" or through the fall. Ozone levels in Sequoia and Kings Canyon are comparable in April and September.**

*As stated in the paper, 90% of days with MD8A > 70 ppb occur during $O_3$ season, with just 10% occurring in spring (2001–2012). Over 2001–2012, we calculate mean MD8A of 51 ppb in April and 76 ppb in September. Therefore, we opt to keep "summer" in the text.*

*We did not say $PO_3$ is only dependent on temperature, just that $PO_3$ is temperature dependent. We have rewritten the sentence to read:*

Page 5, Lines 2–4: "High $O_3$ days are most frequent in SNP and the SJV in the summer through early fall (Pusede and Cohen, 2012; Meyer and Esperanza, 2016), as $PO_3$ chemistry is often temperature-dependent (reviewed in Pusede et al., 2015) and this effect is particularly strong in the SJV (Pusede and Cohen, 2012; Pusede et al., 2014)."

**13) P5, L4: "due to" is inappropriate here – it is a result of the Mediterranean climate.**

*We have changed "due to" to "because of."*

**14) P5, L21: First off, there isn't a methods or experimental section – more information needs to be provided. This get to the more important point that you state that all data are provided by CARB, when in fact, they are not. The data are served up by CARB on their website, but the NPS data is provided by the NPS on their data page, which also gets uploaded to AQS, which is the main repository that houses all national ozone data – it is the single main repository. CARB either serves up the data directly from AQS or in their own database, where they have either obtained the data directly from the NPS or AQS. So, it should be clearly stated who is the proprietor of the data and where it was obtained from. These have been merged into one item, and what is disseminated in this sentence is incorrect.**

*While there is not a dedicated methods section, we believe we have provided readers will all relevant details on which data were used, where data were acquired, and how the calculations were done. We have adjusted the sentence to read:*

Page 5, Lines 26–28: "The data are collected by various agencies, including the National Park Service, and are hosted by the California Air Resources Board and available for download at https://www.arb.ca.gov/aqmis2/aqdselect.php."

**15) P7, L15: The NAAQS for ozone is an 8 hour average value; it is the annual 4th highest daily maximum 8 hour average ozone concentration, averaged over 3 consecutive years (the design value) – this must not exceed 70 ppb. So, what you are saying is repetitive and not conveyed properly. What you should say is the annual 4th highest daily maximum 8 hour average, or the DM8HA or DM8A, not the "8-h O3 NAAQS".**

*We have added a definition of non-attainment, changed NAAQS to MD8A $O_3$ throughout the text, and defined our use of exceedance.*

Page 7, Lines 32–33; Page 8, Line 1: "The MD8A $O_3$ is a human health-based metric computed as the maximum unweighted daily 8-h average $O_3$ mixing ratio. A region is classified as in nonattainment of the NAAQS when the fourth-highest MD8A $O_3$ over a 3-yr period, known as the design value, exceeds a given

standard. In this work, we utilize the seasonal mean MD8A and discuss $O_3$ exceedances as individual days in which MD8A $O_3 > 70.4$ ppb, the current 8-h NAAQS."

**16) P7, L16: Why are trends only reported as a percent change? It would be more useful to include the ppb per yr trend in the table, along with the percent change. Moreover, why is only the 8-hr daily max being listed? I'm assuming it's the annual 4th high daily max 8-hr average, but it's not stated in the text. Please clarify.**

*We have updated the table to include the change in $O_3$ amount per year derived from the slope of the regression. We have clarified the meaning of the MD8A trend in the text, which is the seasonal mean MD8A and not the design value. See previous comment.*

"**Table 1.** $O_3$ changes in Visalia, SEQ1, and SEQ2 over 2001–2012 according to MD8A, SUM0, W126, and morning $O_x$ metrics based on a linear fit of annual mean data (shown in Figure 6) in the springtime and $O_3$ season. Each left column is the percent change with respect to fit value in 2001 at SEQ1 during $O_3$ season for comparison, which is the highest $O_3$ observed for each metric. Each right column is the fit slope with slope errors in $O_3$ abundance units per year."

| $O_3$ metric | MD8A | | SUM0 | | W126 | | Morning $O_x$ | |
|---|---|---|---|---|---|---|---|---|
| | **%** | **ppb y$^{-1}$** | **%** | **ppm h y$^{-1}$** | **%** | **ppm h y$^{-1}$** | **%** | **ppb y$^{-1}$** |
| **$O_3$ season (June–October)** | | | | | | | | |
| SEQ2 | −19 | −1.4 ± 0.41 | −15 | −1.2 ± 0.46 | −37 | −2.2 ± 0.72 | −17 | −1.0 ± 0.32 |
| SEQ1 | −13 | −1.0 ± 0.27 | −12 | −0.96 ± 0.21 | −28 | −1.7 ± 0.36 | −14 | −0.83 ± 0.21 |
| Visalia | −7 | −0.54 ± 0.30 | −3 | −0.20 ± 0.28 | −11 | −0.69 ± 0.41 | −6 | −0.50 ± 0.30 |
| **Springtime (April–May)** | | | | | | | | |
| | **%** | **ppb y$^{-1}$** | **%** | **ppm h y$^{-1}$** | **%** | **ppm h y$^{-1}$** | **%** | **ppb y$^{-1}$** |
| SEQ2 | −13 | −1.0 ± 0.38 | −16 | −1.2 ± 0.47 | −30 | −1.8 ± 0.62 | −13 | −0.78 ± 0.34 |
| SEQ1 | −8 | −0.59 ± 0.42 | −6 | −0.50 ± 0.53 | −24 | −1.5 ± 0.62 | −6 | −0.35 ± 0.32 |
| Visalia | −3 | −0.23 ± 0.39 | −4 | −0.31 ± 0.38 | −11 | −0.69 ± 0.49 | −8 | −0.39 ± 0.35 |

**17) P7, L16: This sentence is not correct – see previous comment. "The 8-h O3 NAAQS is a human health-based metric computed as the maximum unweighted daily 8-h average O3 mixing ratio." As for SUM0 – you need to say why it's called SUM0 – as in, why is it a "0"?**

*We have added a definition of non-attainment, changed NAAQS to MD8A $O_3$ throughout the text, and defined our use of exceedance.*

Page 7, Lines 32–33; Page 8, Line 1: "The MD8A $O_3$ is a human health-based metric computed as the maximum unweighted daily 8-h average $O_3$ mixing ratio. A region is classified as in nonattainment of the NAAQS when the fourth-highest MD8A $O_3$ over a 3-yr period, known as the design value, exceeds a given standard. In this work, we utilize the seasonal mean MD8A and discuss $O_3$ exceedances as individual days in which MD8A $O_3$ > 70.4 ppb, the current 8-h NAAQS."

*We have added clarification for the SUM0 metric:*

Page 8, Lines 1–3: "SUM0 is equal to the sum of hourly $O_3$ concentrations over a 12-h daylight period (8 am–8 pm LT), as opposed to SUM06, which is limited to hourly $O_3$ mixing ratios greater than 60 ppb."

**18) P8, L1-11: For this paragraph, only percentages are reported – it is absolutely necessary to include what the corresponding values were on ppb and ppm hrs for W126 and SUM0. For this to have value to ecosystem and plant effects folks, numbers, not percentages, are needed.**

*The new Figure 6 does this and we have added numbers for each metric at SEQ1 to the text. See comment 5.*

Page 9, Lines 2–11: "For context in SEQ1, during $O_3$ season the mean MD8A declined from 82.3 ppb (2001–2002) to 73.8 ppb (2011–2012), but in the springtime the MD8A fell from 61.7 ppb (2001–2002) to 55.6 ppb (2011–2012). SUM0 $O_3$ fell from 87.0 ppm h (2001–2002) to 79.0 ppm h (2011–2012) during $O_3$ season and from 69.9 ppm h (2001–2002) to 61.8 ppm h (2011–2012) in the springtime. W126 $O_3$ decreased from 67.8 ppm h (2001–2002) to 53.7 ppm h (2011–2012) during $O_3$ season and from 39.8 ppm h (2001–2002) to 25.4 ppm h (2011–2012) in springtime. Morning $O_3$ fell from 67.1 ppb (2001–2002) to 59.6 ppb (2011–2012) during $O_3$ season and from 49.0 ppb (2001–2002) to 45.1 ppb (2011–2012). This pattern was not observed in one instance: SUM0 in SEQ2. Here, seasonal differences were comparable; however, mean daily indices were observed to differ, where SUM0 $O_3$ decreased from 0.914 ppm h (2001–2002) to 0.816 ppm h (2011–2011) during $O_3$ season, and, in the springtime, fell from 0.673 ppm h (2001–2002) to 0.616 ppm h (2011–2012), which amount to a change of –11% during $O_3$ season –8% in the springtime."

**19) Section 3.4 You state that you "predict future $O_3$ levels in the context of protective threshold"; however, it is not stated how your do this in this section – please provide necessary information and figures.**

*We have clarified this point as follows:*

Page 9, Lines 20–34; Page 10, Lines 1–4: "High $O_3$, as defined by exceedances of protective thresholds, also became less frequent over the 12-yr record. The number of days in which MD8A $O_3$ was greater than 70.4 ppb in 2001–2002 (averages are rounded up) was 68 $yr^{-1}$ ($O_3$ season) and 15 $yr^{-1}$ (springtime) in Visalia. In 2011–2012, the number of exceedances fell to 42 $yr^{-1}$ ($O_3$ season) and 6 $yr^{-1}$ (springtime). At SEQ1 in 2001–2002, there were 121 exceedance days $yr^{-1}$ ($O_3$ season) and 21 $yr^{-1}$ (springtime), declining in 2011–2012 to 99 $yr^{-1}$ ($O_3$ season) and 10 $yr^{-1}$ (springtime). At SEQ2 in 2001–2002, there were 103 exceedance days $yr^{-1}$ ($O_3$ season) and 13 $yr^{-1}$ (springtime). In 2011–2012, this decreased to 63 exceedance days $yr^{-1}$ ($O_3$ season) and 3 $yr^{-1}$ in 2011–2012 (springtime).

While there is no standard for SUM0, there are three time-integrated W126 protective thresholds. These are: 5–9 ppm h to protect against visible foliar injury to natural ecosystems, 7–13 ppm h to protect against growth effects to tree seedlings in natural forest stands, and 9–14 ppm h to protect against growth effects to tree seedlings in plantations, known as the 5, 7, and 9 ppm h standards (Heck and Cowling 1997). Rather than calculate W126 exceedances using a 3-month summation of monthly indices, we instead count the number of days required for an exceedance to occur, summing daily W126 indices from the first day of the springtime (1 April). A larger number of days indicates improved air quality. We do this to generate information in addition to exceedance frequency, as W126 $O_3$ at SEQ1 and SEQ2 is greater than all three standards in all years in both seasons. We only consider springtime, as this is when W126 is reported to better correlate with plant $O_3$ uptake (Panek et al., 2002; Kurpius et al., 2002; Bauer et al., 2000). At SEQ1 from 1 April in 2001–2002, 37, 41, and 45 days of $O_3$ accumulation reached exceedances of the 5, 7, and 9 ppm h thresholds, respectively (averages are rounded up). In 2011–2012, 3 to 13 more days were needed at SEQ1, as 40, 49, and 58 days of $O_3$ accumulation were required to exceed the 5, 7, and 9 ppm h thresholds. At SEQ2 from 1 April in 2001–2002, 41, 46, and 49 days of accumulation led to exceedance of the 5, 7, and 9 ppm h thresholds, respectively. In 2011–2012, 59, 65, and 73 days were required at SEQ2, or 18–24 more days."

**20) "Future exceedances are computed assuming individual daily indices continue to decline at the 2001–2012 rate and are projected from 2011–2012 values." Is this a reasonable assumption? I'm not convinced this is the case. There is ozone data beyond 2012 at these sites, so does the rate of decline hold true? The fact that you are predicting future ozone levels off of this would suggest it should be evaluated.**

*We have removed our projections of future exceedances and instead included a discussion of Val Martin et al. (2015) and known regulations.*

Page 11, Lines 21–34; Page 12, Lines 1–2: "Because $PO_3$ in SNP is $NO_x$-limited, future $NO_x$ reductions are expected to have at least as large an impact on local $PO_3$ as past reductions. Seasonal mean $NO_2$ concentrations have decreased by 58% and 53% in Visalia in springtime and $O_3$ season, respectively. Local $NO_x$ emissions should continue to decline into the future, as there are significant controls currently ongoing or in the implementation phase, including more stringent national rules on heavy-duty diesel engines (Environmental Protection Agency, 2000), combined with California Air Resources Board (CARB) diesel engine retrofit-replacement requirements (California Air Resources Board, 2008), and more stringent CARB standards for gasoline-powered vehicles (California Air Resources Board, 2012). While $O_3$ declines near or greater than those that occurred from 2001 to 2012 are required to eliminate exceedances in SNP, modeling analysis by Lapina et al. (2014) suggests that W126 in the region would be well below these thresholds in the absence of anthropogenic precursor emissions, implying further emissions controls would be effective. Under the stringent precursor controls of RCP4.5, Val Martin et al. (2015) projected decreases of 11% and 67% for the MD8A and W126 in 2050, respectively, from the base year of 2000, with mean $O_3$ decreasing from 58.9 ppb (MD8A) and 45.5 ppm h (W126) in 2000 to 52.7 ppb (MD8A) and 15.1 ppm h (W126). Under the RCP8.5, smaller $O_3$ declines were predicted, with MD8A unchanged and W126 falling by 38% to 28.3 ppm h. Given that these scenarios represent a reasonable spread of possible future climatic conditions, Val Martin et al. (2015) suggest at least W126 will remain well above protective thresholds in 2050."

**21) "If past decreases in O3 continue over the next two decades, we predict no exceedances of the 8-h O3 NAAQS at SEQ2 by 2021 in springtime and by 2031 during O3 season, no exceedance of the 9-ppm h W126 threshold by 2021, and no further exceedances of 5- and 7-ppm h thresholds by 2031."**

**Following suit, more information needs to be provided to make such a bold statement when using such a rudimentary method. How much is NOx going to go down? How are large scale circulation patterns (e.g., PDO, ENSO, etc.) going to change and influence what is being transported in? What about the different climate futures? There are an array of emissions scenarios that can lead to significant differences in what you are inaccurately and inappropriately conveying here. Also, climate change - This section either must be expanded upon significantly or simply removed from the paper. As an example that contradicts your statements about ozone exceedances, the following is pulled directly from Val Martin et al. (2015) for Sequoia and Kings Canyon. Here, the report the actual values using different climate futures in order to assess what the ozone and W126 values will be. According to their rigorous analysis, both ozone and W126**

**values will exceed the current NAAQS level of 70 ppb and W126 values will also increase, and be well above the 5-9 ppm hr range.**

**Summer MDA-8 Ozone (ppb)**

**2000 (Baseline): 71.3, 2050 (RCP4.5): 72.9, 2050 (RCP8.5): 73.8**

**O3 W126 (ppm-hr)**

**2000 (Baseline): 46.0, 2050 (RCP4.5): 50.6, 2050 (RCP8.5): 53.2**

**Val Martin, M., C. L. Heald, J.-F. Lamarque, S. Tilmes, L. K. Emmons, and B. A. Schichtel**

**How emissions, climate, and land use change will impact mid-century air quality over the United States: a focus on effects at national parks, Atmos. Chem. Phys., 15, 2805–2823, 2015, www.atmos-chem-phys.net/15/2805/2015/.**

*We thank the reviewer for the reference. We have included discussion of Val Martin et al. (2015) as shown below. However, the quoted numbers must come from another article. Val Martin et al. (2015) report the following for Sequoia, which do imply O₃ declines:*

*Summer MDA-8 Ozone (ppb)*

*2000 (Baseline): 58.9, 2050 (RCP4.5): 52.7, 2050 (RCP8.5): 58.9*

*O3 W126 (ppm-hr)*

*2000 (Baseline): 45.5, 2050 (RCP4.5): 15.1, 2050 (RCP8.5): 28.3*

Page 10, Lines 26–31: "With the Community Earth System Model, Val Martin et al. (2015) modeled air quality in national parks under two Representative Concentration Pathway (RCP) scenarios, computing substantially larger decreases over a 50-yr period in W126 $O_3$ compared to the MD8A. Considering that the SUM0 metric has been shown to best correspond to plant $O_3$ uptake in Sierra Nevada forests using $O_3$ flux observations (Panek et al., 2002) and that we observe W126 $O_3$ has declined at approximately twice the rate of SUM0 over 2001–2012, W126 trends may provide an overly optimistic representation of past declines in ecosystem $O_3$ impacts in SNP."

**22) P9, L30: "O₃ reductions predicted by W126 are almost twice those of SUM0." What does this statement mean? How are ozone reductions predicted by W126 or SUM0? Both of these metrics are determined from ozone levels – how are these used to predict ozone reductions?**

*Changed as follows:*

Page 10, Line 23: "Reductions in ecosystem $O_3$ impacts as represented by declines in W126 are greater than those of SUM0."

Page 13, Line 3; Page 14, Lines 1–2: "$O_3$ decreases over 2001–2012 computed with W126 are almost double those for SUM0, with the W126 emphasis of higher $O_3$ concentrations giving the most optimistic evaluation of the efficacy of past emission controls."

**23) P10,L2: Regarding the following statement: ". . .W126 likely provides an overly optimistic representation of past and future trends in O3 impacts in SNP.", this is a rather bold statement to make to summarize the paragraph, yet you provide no hard evidence of this – there is nothing in this section that supports this statement. Please address this in a more rigorous manner.**

*We clarified our logic. Greater decreases in W126 relative to other $O_3$ metrics have also been reported by two national parks-focused modelling studies: Lapina et al. (2014), as we mentioned in the initial submission, and Val Martin et al. (2015). We have added Val Martin et al. (2015) to the discussion on this point.*

Page 10, Lines 23–31: "Reductions in ecosystem $O_3$ impacts as represented by declines in W126 are greater than those of SUM0. We attribute this difference to the W126 weighting algorithm that makes the metric most sensitive to changes in the highest $O_3$. Using the GEOS-Chem model with a focus on national parks, Lapina et al. (2014) also found W126 was more responsive to decreases in anthropogenic emissions than daily (8 am–7 pm, LT) average $O_3$ concentrations. With the Community Earth System Model, Val Martin et al. (2015) modeled air quality in national parks under two Representative Concentration Pathway (RCP) scenarios, computing substantially larger decreases over a 50-yr period in W126 $O_3$ compared to the MD8A. Considering that the SUM0 metric has been shown to best correspond to plant $O_3$ uptake in Sierra Nevada forests using $O_3$ flux observations (Panek et al., 2002) and that we observe W126 $O_3$ has declined at approximately twice the rate of SUM0 over 2001–2012, W126 trends may provide an overly optimistic representation of past declines in ecosystem $O_3$ impacts in SNP."

**24) P10, L9: For the following statement: ". . .leading to policies not optimized to decrease O₃ in cooler springtime conditions." Please elaborate on this point - this needs to be shown quantitatively. How large or small of a difference are you suggesting? What are the policies? How are they not optimized for the cooler springtime conditions? What could/should be done to address this policies in order to optimize them for the springtime?**

*We have elaborated on why policies may not be optimized for springtime and given two examples of what the results of this might be. Without performing model simulations, we cannot quantify these effects, but we have widened the discussion to be more specific and we believe more useful. The new text is included in response to comment 3 and shown below:*

Page 12, Lines 3–23: "Over 2001–2012, $O_3$ declines have mostly been smaller in SNP when plant $O_3$ uptake is greatest (springtime), despite comparable $NO_x$ decreases in both seasons. This may be in part because regulatory strategies prioritize attainment of the $O_3$ NAAQS in polluted urban areas like the SJV basin, where air parcels influenced by the results of these controls are then transported downwind to locations with different $PO_3$ chemistry. In the development of regulatory plans, agencies use models to hindcast past $O_3$ episodes, facilitating testing of the efficacy of specific $NO_x$ and/or organic emissions reductions over that episode to meet the 8-h $O_3$ NAAQS or progress goals (Environmental Protection Agency, 2007; Environmental Protection Agency, 2014). In nonattainment areas, U.S. EPA guidance recommends modeling past time periods that meet a number of specific criteria, such as typifying the meteorological conditions that correspond to high $O_3$ days as defined by the MD8A greater than the NAAQS value and focusing on the ten highest modeled $O_3$ days (Environmental Protection Agency, 2007; Environmental Protection Agency, 2014). Regulatory modeling in the SJV (Visalia, SEQ1, and SEQ2 are included in this attainment demonstration) is more comprehensive, as it was recently updated to span the full $O_3$ season (defined as May–September); still potential reductions (known as relative reduction factors, RRFs) are based on the MD8A and restricted to high $O_3$ days (San Joaquin Valley Air Pollution Control District, 2007; San Joaquin Valley Air Pollution Control District, 2014). In the SJV, high $O_3$ days are most frequent in the late summer ($O_3$ season) and on the hottest days of the year (Pusede and Cohen, 2012). Even in SEQ1 and SEQ2, days with MD8A > 70.4 ppb are far more common in the summer. Because of chemical and meteorological differences between seasons, this may lead to policies not optimized to decrease $O_3$ in cooler springtime conditions, which in the SJV are more $NO_x$-suppressed and therefore more sensitive to controls on reactive organic compounds (Pusede et al., 2014). In addition, we observe greater year-to-year $O_3$ variability in the springtime than during $O_3$ season (Figure 6), suggestive of a larger relative role of interannual meteorological variability controlling $O_3$. Deeper cuts in emissions would be required in the springtime, as decreases in anthropogenic emissions have a proportionally smaller effect on the total $O_3$ abundance than during $O_3$ season."

**25) P10,L15: Regarding the following "Third, aircraft observations collected in the direction of daytime upslope flow from the SJV to Sierra Nevada foothills reveal substantial decreases in NOx concentrations relative to isoprene, a key contributor to**

total organic reactivity (e.g., Beaver et al., 2012)." You are consding the 2001-2012 time frame, how representative is this single day? Can this be put in to greater context?

*We have added this text:*

Page 6, Lines 26–31: "While these data were collected on one day in a different year from our study, the

5      relative pattern of $NO_x$ to organic compound emissions is likely representative, as there have been no substantial changes in the locations of urban $NO_x$ and biogenic organic emitters. This $NO_x$ to organic compound gradient is consistent with observations over longer sampling periods downwind of the Central California city of Sacramento, where the $NO_x$-enriched Sacramento urban plume is transported up the western slope of the vegetated Sierra Nevada Mountains (e.g., Beaver et al., 2012; Dillion et la., 2002;

10      Murphy et al., 2006)."

**26) P10, L18: For the following sentence: "This implies that $PO_3$ in Visalia and SNP is differently sensitive to emission controls, with SNP more responsive to $NO_x$ emissions control than Visalia." This is only one aspect of the issue, the other is that you are sitting in a source region, so the regime you are in is different; also, there is mixing and**

15 **dilution that occur with transport, so this is another major factor - it's not simply response to emissions controls. This needs to be addressed and put into context.**

*We have changed the text as follows:*

Page 11, Lines 11–14: "Distinct local $PO_3$ regimes lead to $PO_3$ chemistry in Visalia and SNP that is differently sensitive to emission controls, with $NO_x$-limited SNP historically more responsive to $NO_x$

20      emission control than Visalia. SNP $NO_x$-limitation is enhanced by $NO_x$ dilution during transport, which further decreases $NO_x$ relative to the abundance of local organic compounds."

**27) P11, L15-16: ". . .day due the mixing. . ." please fix this sentence, and it would be best not to use due to. . .**

*Corrected:*

25      Page 13, Line 31: "…which results from the mixing…"

**28) As it currently stands, the data disseminated in the tables is not very useful, especially Table 2. What would be better to provide in Table 2 are the projected DM8HA values in ppb and the W126 values in ppm hrs, along with their corresponding #s of exceedances per year. However, the method used for this work is not suitable for**

30 **providing any type of reasonable predicted value. As for Table 1, actual values should be included along with the percent change.**

*Table 1 has been updated and Figure 6 added to show actual values. Table 2 has been deleted.*

**In summary, before this paper is worthy of being published, there are significant issues that must be addressed.**

---

## Author Response (AR2)

**Author response**
On the effect of upwind emission controls on ozone in Sequoia National Park

**Referee Comments (bold)**

*Author Response (italics)*

Author Changes to Manuscript (standard)

We thank the reviewer for their feedback, which has improved the quality of the manuscript. We address each point below.

**Anonymous Referee #3**

**Buysee et al provide an analysis of ozone data and trends in Sequoia National Park and in the upwind source area of the San Joaquin Valley represented by a site in Visalia, CA. They focus their analysis on the relationship between these two areas and on a comparison of metrics typically used for human health assessments versus metrics relevant to vegetation effects. This is a useful exploration of ozone in an area where ozone is expected to have substantial impacts on vegetation and ecosystem health especially since relatively few ambient ozone analyses in the literature focus on ecosystem impacts. The paper is generally well written and the data analysis is sound. There are some improvements that could be made in terms of background information and references. In addition, there are some places where minor clarifications/updates to the analysis and presentation of the results would be beneficial. I recommend the paper be published after the following comments are addressed.**

**Introduction:**

**1) Lines 28-30: I think this comparison is misleading for a few reasons. First, as the authors clearly understand, ozone is not highest near emissions sources but rather downwind within a metro area. While the number of days above 70 ppb in LA proper may be 76, the number of days above 70 in the LA metro area (defined as the Los Angeles Nonattainment area which includes portions of Riverside and San Bernardino) is actually in the range of 150-175. I think that is a more appropriate comparison, otherwise the authors give the false impression that SNP experiences more days above 70 ppb than the LA metro area. In addition, while determinations of the three most ozone polluted cities in the US by the ALA is likely based on the US standard of the 4th highest MD8A value, the number of days above a threshold has a lot to do with regional climate. Therefore, it would be more appropriate to compare SNP to other local urban areas (such as those listed on line 14 of page 4) than to Denver and Phoenix. Also note**

**that the number of days above 70 ppb in Phoenix has decreased dramatically over the time-period evaluated (more than 100 days in 2001).**

*Deleted.*

**2) The recent Tropospheric Ozone Assessment Report (TOAR) presented analysis that is relevant to the work described here. There are two articles of particular significance: Lefohn et al (2018) describe the relationship between various ozone health and vegetation metrics and show how trends in different metrics compare across the US, EU and Asia. Mills et al (2018) specifically focus on characterizing ozone levels and trends for metrics most important to vegetation health. These articles should be discussed in the introduction. In addition, it would be worthwhile to put your results in context of those other recent findings both within California and globally.**

*We thank the reviewer for the references, which were published after our initial submission in October 2017. We have added a discussion of findings from TOAR as follows:*

Page 2, Lines 32–34: "While trends toward median $O_3$ levels observed at a large number of U.S. sites (Lefohn et al., 2017) may have decreased the number of NAAQS exceedances, benefits to plants and ecosystems may be limited (Lefohn et al., 2018)."

Page 3, Lines 9–11: "For a global assessment of $O_3$ distribution and trends using a variety of ecosystem concentration metrics see Mills et al. (2018), which is part of the Total Ozone Assessment Report (TOAR)."

Page 10, Lines 31–32; Page 11, Lines 1–2: "This is particularly true in Mediterranean ecosystems like SNP and under drought conditions (e.g., Panek et al., 2002), which is where and when plant $O_3$ uptake and high atmospheric $O_3$ concentrations may be uncorrelated. This may also be true in European Mediterranean climate regions, where high concentrations of ecosystem-based $O_3$ metrics have also been observed (Mills et al., 2018)."

Page 11, Lines 20–22: "In the TOAR global analysis, Mills et al. (2018) found April–September W126 downward trends over 1995–2014 in California of between 1–2 ppm h $yr^{-1}$ to be among the most rapid W126 declines in the world."

Page 9, Lines 5–8: "Table 2 coloration indicates trend significance computed using the Mann-Kendall non-parametric test following the categorization developed by TOAR authors (Chang et al., 2017; Mills et al., 2018; Lefohn et al., 2018), with p-values categorized as statistically significant (0–0.05), indicative of a trend (0.05–0.10), a weak indication of change (0.10–0.34), and weak or no change (0.34–1)."

**3) EPA, 2015 would be a more appropriate reference for statements at the top of page 3 than EPA, 2016 and EPA, 2010. Also, EPA, 2010 is not listed in the references section.**

*We thank the reviewer for catching this error. We have added the EPA, 2010 reference to the references section. We have also adjusted the first EPA reference to cite EPA, 2015. We retain the subsequent EPA, 2010 reference, which includes the proposal of the weighted W126 metric as the secondary $O_3$ standard, as well as the EPA, 2016 reference, which provides specific detail on the W126 metric.*

Page 2, Lines 28–34; Page 3, Lines 1–3: "While there is a secondary NAAQS requirement aimed at vegetation protection, this has historically been the same metric (based on the MD8A $O_3$) at the same threshold as the primary NAAQS (Environmental Protection Agency, 2015b). Plants and ecosystems have been shown to be sensitive to lower $O_3$ concentrations, over longer-term exposures, and at different times of day and year than when NAAQS exceedances are frequent (e.g., Kurpius et al., 2002; Panek 2004; Panek and Ustin, 2005; Fares et al., 2013). While trends toward median $O_3$ levels observed at a large number of U.S. sites (Lefohn et al., 2017) may have decreased the number of NAAQS exceedances, benefits to plants and ecosystems may be limited (Lefohn et al., 2018). The U.S. Environmental Protection Agency (EPA) has considered redefining the secondary standard to reflect ecological systems, with the W126 metric put forth (Environmental Protection Agency, 2010). W126 is a 12-h daily 3-month summation weighted to emphasize higher $O_3$ concentrations (Environmental Protection Agency, 2006; Environmental Protection Agency, 2016) that is used by the U.S. National Park Service."

**4) Line 9, Page 3: Is SUM0 the same as the M12 metric from the TOAR analysis?**

*The M12 metric is an average exposure metric, while SUM0 is a cumulative exposure metric. Since both are over the same time window (8 am–8 pm LT), they represent the same concentrations when computed over a 3-month period. In this work, we retain the cumulative exposure framework that aligns with SUM06 and W126 metrics, which have been explicitly compared to $O_3$ eddy-covariance flux measurements in Sierra Nevada forests.*

Page 3, Lines 7–9: "Similarly, the M12 metric is an average exposure metric computed over the same daily time window (8 am–8 pm LT) and representing the same hourly $O_3$ concentrations when computed over a 3-month period as SUM0 (e.g., Tingey et al., 1991; Lefohn and Foley, 1993)."

**Methods:**

**1) Much of the methods description appears to be in the results section with bits included in the introduction. I think the article would be clearer if the methods were broken out into a separate section. The methods section could include: description of monitoring sites (currently lines 23-29 on page 5), description of where data were retrieved from, gap filling methodology (currently lines 10-14 on page 8), calculation of metrics (currently mentioned in various places in the introduction and results section), and description of trends calculation methodology (not currently included in the text at all).**

*We have opted for the referee's second suggestion and moved the measurement description to the general results section (section 3). We have added a description of the trend and significance calculation methodology, referencing the TOAR reports.*

Page 5, Lines 24–29: "Hourly $O_3$ data have been routinely collected in SNP at two monitoring stations, a lower elevation site, SNP-Ash Mountain (36.489 N, 118.829 W), at 515 m above sea level (ASL) and a higher elevation site, SNP-Lower Kaweah (36.566 N, 118.778 W), at 1926 m ASL (Figure 1). We refer to these stations as SEQ1 and SEQ2, respectively. $O_3$ and $NO_2$ data are measured in Visalia (36.333 N, 119.291 W), which is in the upwind direction of SNP at 102 m ASL (Figure 2). The data are collected by various agencies, including the National Park Service, and are hosted by the California Air Resources Board and available for download at https://www.arb.ca.gov/aqmis2/aqdselect.php."

Page 9, Lines 1–2: "In Figure 6, mean seasonal daily MD8A and morning metrics and cumulative SUM0 and W126 metrics are shown for Visalia, SEQ1, and SEQ2 with their fit derived using an ordinary least squares linear regression."

**2) Why did analysis only use data through 2012? Certified ozone data is available through 2017. I suggest that the authors extend their analysis to include more recent data.**

*California has recently experienced the worst drought in recorded history. Because this manuscript focuses on the effects of upwind emission controls on $O_3$, we stopped the analysis before this severe drought period. We have added text to this point.*

Page 3, Lines 29–33: "In this paper, we report $O_3$ trends from 2001 to 2012 in SNP and the upwind SJV city of Visalia to study the effects of SJV emission controls on SNP $O_3$. We do not extend the analyses beyond 2012 as, beginning in late 2012, California experienced the worst drought in recorded history (Griffin and

Anchukaitis, 2014; Diffenbaugh et al., 2015). Because $O_3$ concentrations are influenced by drought conditions (e.g., Jacob and Winner, 2009; Huang et al., 2016), we focus on the 2001 to 2012 time period."

**3) What methods did the authors use to calculate trends? There are many different linear regression and other methods that could be used to determine trend magnitudes and significance. For instance, other trends analysis have used ordinary least squares regressions, Thiel-Sen regressions, Spearman rank order regression etc. Some regression methods also account for temporal variability on different time scales (inter-annual variability versus seasonal variability etc). Please specify the method used here. Also, please add p-values to the trends magnitudes reported in Table 1 and include some description of which trends were statistically significant in the results section.**

*We have stated our regression methodology and indicated which trends were statistically significantly different than the null hypothesis (i.e. no trend).*

Page 9, Lines 1–2: "In Figure 6, mean seasonal daily MD8A and morning metrics and cumulative SUM0 and W126 metrics are shown for Visalia, SEQ1, and SEQ2 with their fit derived using an ordinary least squares linear regression."

Page 9, Lines 5–8: "Table 2 coloration indicates trend significance computed using the Mann-Kendall non-parametric test following the categorization developed by TOAR authors (Chang et al., 2017; Mills et al., 2018; Lefohn et al., 2018), with p-values categorized as statistically significant (0–0.05), indicative of a trend (0.05–0.10), a weak indication of change (0.10–0.34), and weak or no change (0.34–1)."

Page 9, Lines 30–34: "This site-dependence is reflected in the $O_3$ trend p-values (Table 2), where at SEQ2, slopes are either statistically significant at the 0.05 level or indicative of a trend (0.05–0.10) for each metric in both seasons. At SEQ1, slopes are statistically significant during $O_3$ season, but only indicative (W126), weakly indicative (MD8A), or suggestive of weak to no change (SUM0 and Morning $O_x$) in springtime. Trends in Visalia are the least robust, with p-values typically only weakly indicative or suggestive of minor to no change in both springtime and $O_3$ season."

**Table 2.** $O_3$ changes in Visalia, SEQ1, and SEQ2 over 2001–2012 according to MD8A, SUM0, W126, and morning $O_x$ metrics based on a linear fit of annual mean data (shown in Figure 6) in the springtime and $O_3$ season. Each left column is the percent change with respect to fit value in 2001 at SEQ1 during $O_3$ season for comparison, which is the highest $O_3$ observed for each metric. Each right column is the fit slope with slope errors in $O_3$ abundance units per year. Coloration is based on the TOAR categorization for trend significance (Lefohn et al., 2018), with p-values calculated using the Mann-Kendell non-parametric test: yellow, 0–0.05, statistically significant trend; green, 0.05–0.10, indicative of a trend; violet, 0.10–0.034, weak indication of change; and pink, 0.34–1, weak or no change.

| O$_3$ metric | MD8A | | SUM0 | | W126 | | Morning O$_x$ | |
|---|---|---|---|---|---|---|---|---|
| | **%** | **ppb yr$^{-1}$** | **%** | **ppm h yr$^{-1}$** | **%** | **ppm h yr$^{-1}$** | **%** | **ppb yr$^{-1}$** |
| | | | **O$_3$ season (June–October)** | | | | | |
| SEQ2 | –19 | –1.4 ± 0.41 | –15 | –1.2 ± 0.46 | –37 | –2.2 ± 0.72 | –17 | –1.0 ± 0.32 |
| SEQ1 | –13 | –1.0 ± 0.27 | –12 | –0.96 ± 0.21 | –28 | –1.7 ± 0.36 | –14 | –0.83 ± 0.21 |
| Visalia | –7 | –0.54 ± 0.30 | –3 | –0.20 ± 0.28 | –11 | –0.69 ± 0.41 | –6 | –0.50 ± 0.30 |
| | | | **Springtime (April–May)** | | | | | |
| | **%** | **ppb yr$^{-1}$** | **%** | **ppm h yr$^{-1}$** | **%** | **ppm h yr$^{-1}$** | **%** | **ppb yr$^{-1}$** |
| SEQ2 | –13 | –1.0 ± 0.38 | –16 | –1.2 ± 0.47 | –30 | –1.8 ± 0.62 | –13 | –0.78 ± 0.34 |
| SEQ1 | –8 | –0.59 ± 0.42 | –6 | –0.50 ± 0.53 | –24 | –1.5 ± 0.62 | –6 | –0.35 ± 0.32 |
| Visalia | –3 | –0.23 ± 0.39 | –4 | –0.31 ± 0.38 | –11 | –0.69 ± 0.49 | –8 | –0.39 ± 0.35 |

**4) Metrics: I suggest that the authors also include an analysis of the AOT40 metric which is commonly used in Europe and is mentioned in the introduction.**

*We have opted to use SUM0 because it has been shown to best correlate with springtime stomatal O$_3$ uptake in a Sierra Nevada forest according to O$_3$ eddy-covariance flux measurements (Panek et al., 2002; Fares et al., 2010). We used W126 because it is a standard metric of the U.S. National Park Service. Past O$_3$ eddy-covariance observations indicate poor correlation between AOT40 and O$_3$ uptake in both the springtime and summer at a Sierra Nevada forest measurement site (Fares et al., 2010). For this reason, we decided to exclude trends in the AOT40 metric. We have added text to this clarify our decision.*

Page 8, Lines 1–4: "In Figure 6 and Table 2, we report 12-yr O$_3$ trends (2001–2012) in SNP and the SJV in springtime and during O$_3$ season using four concentration metrics: MD8A; two common vegetative-based indices, SUM0 and W126; and a morning average metric. We do not report trends in SUM06 or AOT40 vegetative indices, as they have been shown to poorly correlate with O$_3$ uptake at a Sierra Nevada forest site even in springtime (Panek et al., 2002; Fares et al., 2010b)."

**5) The authors have some limited analysis of number of days above 70 ppb (lines 20-25 on page 9) but they should also calculate trends in this metric similar to the other metrics. These trends for # of days > 70 could be included in both Table 1 and Figure 6.**

*It is our aim discuss trends in the metrics that have been shown to be best correlated to past O$_3$ flux measurements in the region, not to provide a comprehensive survey of trends*

*in all metrics. The TOAR reports have now done this in a more spatially comprehensive fashion. To keep the manuscript focused, we do not add a new figure, but do add the trend and significant information to the text.*

Page 10, Lines 7–10: "Patterns in high MD8A $O_3$ days follow trends in other metrics, with the largest rates of change occurring during $O_3$ season in SEQ2 (–4.7 days $yr^{-1}$, $p = 0.02$), then SEQ1 (–2.8 days $yr^{-1}$, $p = 0.05$) and Visalia (–2.5 days $yr^{-1}$, $p = 0.05$). In springtime, smaller decreases are observed with similar spatial patterns, SEQ2 (–1.0 days $yr^{-1}$, $p = 0.02$), SEQ1 (–1.0 days $yr^{-1}$, $p = 0.15$) and Visalia (–0.5 days $yr^{-1}$, $p = 0.37$)."

**6) The authors use a cutoff of 70.4 ppb for the US ozone standard, but the US EPA actually truncates rather than rounds fractional ppb values when calculating days above the standard. So, the correct threshold should actually be 70.9 ppb. See description in 40 CFR Appendix U to Part 50 from October 2015 (https://www.law.cornell.edu/cfr/text/40/appendix-U_to_part_50)**

*We have adjusted our threshold calculation, corresponding text, and numerical results.*

Page 1, Lines 27–29: "Ozone ($O_3$) concentrations in SNP exceeded the current U.S. human health-based $O_3$ National Ambient Air Quality Standard (NAAQS), defined as maximum daily 8-h average (MD8A) $O_3$ greater than 70 ppb, on an average of 117 days per year over the time period 2001–2012."

Page 8, Lines 7–8: "In this work, we utilize the seasonal mean MD8A and discuss $O_3$ exceedances as individual days in which MD8A $O_3 > 70.9$ ppb, the current 8-h NAAQS (CFR 40, 2015)."

Page 10, Lines 1–6: "High $O_3$, as defined by exceedances of protective thresholds, also became less frequent over the 12-yr record. The number of days in which MD8A $O_3$ was greater than 70.9 ppb in 2001–2002 (averages are rounded up) was 66 $yr^{-1}$ ($O_3$ season) and 14 $yr^{-1}$ (springtime) in Visalia. In 2011–2012, the number of exceedances fell to 39 $yr^{-1}$ ($O_3$ season) and 6 $yr^{-1}$ (springtime). At SEQ1 in 2001–2002, there were 119 exceedance days $yr^{-1}$ ($O_3$ season) and 20 $yr^{-1}$ (springtime), declining in 2011–2012 to 97 $yr^{-1}$ ($O_3$ season) and 10 $yr^{-1}$ (springtime). At SEQ2 in 2001–2002, there were 103 exceedance days $yr^{-1}$ ($O_3$ season) and 13 $yr^{-1}$ (springtime). In 2011–2012, this decreased to 62 exceedance days $yr^{-1}$ ($O_3$ season) and 3 $yr^{-1}$ in 2011–2012 (springtime)."

Page 13, Line 8: "Even in SEQ1 and SEQ2, days with MD8A $> 70.9$ ppb are far more common in the summer."

**Results:**

**1) If the authors decide not to include a separate methods section, then I suggest the authors at least move the general description of monitoring sites (lines 23-28 on page 5) up to the general results section (section 3) instead of having it buried in section 3.1.**

5 *We have moved this description to section 3 as recommended.*

Page 5, Lines 24–29: "Hourly $O_3$ data have been routinely collected in SNP at two monitoring stations, a lower elevation site, SNP-Ash Mountain (36.489 N, 118.829 W), at 515 m above sea level (ASL) and a higher elevation site, SNP-Lower Kaweah (36.566 N, 118.778 W), at 1926 m ASL (Figure 1). We refer to these stations as SEQ1 and SEQ2, respectively. $O_3$ and $NO_2$ data are measured in Visalia (36.333 N, 119.291 W),

10 which is in the upwind direction of SNP at 102 m ASL (Figure 2). The data are collected by various agencies, including the National Park Service, and are hosted by the California Air Resources Board and available for download at https://www.arb.ca.gov/aqmis2/aqdselect.php."

**2) Line 2, page 6: Additionally, rush-hour NOx emissions are likely to impact the diurnal pattern of Ox described here for Visalia.**

15 *We have added text to this point.*

Page 6, Lines 3–7: "In Visalia, $O_x$ concentrations increase sharply beginning in early morning (7 am LT) until 2 pm LT, continuing to rise slightly until 4–5 pm LT (Figure 4). This diurnal pattern reflects a combination of local $PO_3$ (the initial rise) and advection of $O_x$ from the upwind source region (late afternoon maximum). In the morning, enhanced rush-hour $NO_x$ emissions overlap in time with the initial increase in $O_x$ with 30–

20 40% of $O_x$ as $NO_2$ at 7–8 am LT. In the afternoon, from 12–4 pm LT ~10–15% of $O_x$ is $NO_2$. At 5 pm, $NO_2$ concentrations increase with evening rush hour with 30–40% of $O_x$ as $NO_2$ at 5–6 pm LT."

**3) Section 3.2: If this analysis is important enough to warrant its own section, then the authors should include a Table or Figure displaying the results described in lines 16-27 of page 7, so that readers will easily be able to see a summary of these results.**

25 *We have added a table of these results and modified the discussion to reference this table.*

Page 7, Lines 20–31: "At moderate temperatures, statistically significant weekday-weekend differences were observed (Table 1). During $O_3$ season, $O_x$ was 6.3 ± 3.5% higher on weekends than weekdays in Visalia, indicating local $PO_3$ was $NO_x$ suppressed. At the same time, $O_3$ was 4.6 ± 3.3% and 4.9 ± 3.9% higher on weekdays than weekends at SEQ1 and SEQ2, respectively, implying $PO_3$ in SNP was $NO_x$ limited. A similar

30 pattern was observed during springtime, as $O_x$ was 7.4 ± 4.6% higher on weekends than weekdays in Visalia

and $O_3$ was $3.5 \pm 7.4\%$ and $4.7 \pm 5.5\%$ higher on weekdays than weekends in SEQ1 and SEQ2. These weekday-weekend patterns imply that a substantial portion of $O_3$ in SNP is produced by low-$NO_x$ $PO_3$ chemistry during air transport from the SJV. At high temperatures, greater weekday concentrations in $O_x$ in Visalia and $O_3$ at SEQ1 and SEQ2 imply $NO_x$-limited chemistry in all three locations (Table 1). Averaged across sites, percent differences in weekdays and weekends (relative to weekdays) were $8.7 \pm 4.8\%$ in the springtime and $4.3 \pm 2.3\%$ during $O_3$ season. $PO_3$ during upslope transport is not apparent by this method because $PO_3$ was also $NO_x$ limited in Visalia, indicating a portion of $O_3$-forminig organic reactivity in Visalia was temperature dependent, consistent with past analyses in other SJV cities (Steiner et al., 2006; Pusede and Cohen, 2012; Pusede et al., 2014; Rasmussen et al., 2014)."

**Table 1.** Percent difference in afternoon (12–6 pm LT) $O_x$ or $O_3$ on weekdays and weekends calculated as: $(O_{x,weekday} - O_{x,weekend}) / O_{x,weekday}$ in Visalia, SEQ1, and SEQ2 in 2001–2003 on moderate and high temperature days. Errors are reported as standard errors of the mean.

| Temperature | Moderate | High |
|---|---|---|
| **$O_3$ season (June–October) 2001–2003** | | |
| | **%** | **%** |
| SEQ2 | $4.9 \pm 3.9$ | $3.5 \pm 2.4$ |
| SEQ1 | $4.6 \pm 3.3$ | $4.2 \pm 1.9$ |
| Visalia | $-6.3 \pm 3.5$ | $5.3 \pm 2.6$ |
| **Springtime (April–May) 2001–2003** | | |
| | **%** | **%** |
| SEQ2 | $4.7 \pm 5.5$ | $5.2 \pm 4.6$ |
| SEQ1 | $3.5 \pm 7.4$ | $8.6 \pm 4.9$ |
| Visalia | $-7.4 \pm 4.6$ | $12.2 \pm 4.8$ |

**4) Lines 16-20 on page 7: Is it also possible that SNP becomes VOC limited on hot days? Maybe due to changes in biogenic VOC emissions?**

*In locations with substantial biogenic emissions like SNP, increases in temperature drive chemistry to be more $NO_x$-limited. We have stated this more explicitly.*

Page 7, Lines 26–31: "At high temperatures, greater weekday concentrations in $O_x$ in Visalia and $O_3$ at SEQ1 and SEQ2 imply $NO_x$-limited chemistry in all three locations (Table 1). Averaged across sites, percent

differences in weekdays and weekends (relative to weekdays) were 8.7 ± 4.8% in the springtime and 4.3 ± 2.3% during $O_3$ season. $PO_3$ during upslope transport is not apparent by this method because $PO_3$ was also $NO_x$ limited in Visalia, indicating a portion of $O_3$-forminig organic reactivity in Visalia was temperature dependent, consistent with past analyses in other SJV cities (Steiner et al., 2006; Pusede and Cohen, 2012; Pusede et al., 2014; Rasmussen et al., 2014)."

**5) Lines 7-8, page 8: It might be useful to note in the description of W126 that the sigmoidal weighting function has an inflection point at around 60 ppb, so days below 60 ppb receive little weighting while days above 60 ppb are weighted heavily.**

*We have added text to this point.*

Page 8, Lines 14–15: "W126 weighting is sigmoidal, with hourly $O_3$ weights equal to $(1 + 4403e^{-126[O3]})^{-1}$, such that hourly mixing ratios below (above) 60 ppb receive less (more) weight (Environmental Protection Agency, 2016)."

**6) Lines 10-11, page 8: How many months had less than 75% complete data at these three sites?**

*We have added text to this point.*

Page 8, Lines 19–20: "From 2001–2012, 0, 8, and 3 months were initially less than 75% complete in Visalia, SEQ1, and SEQ2, respectively."

**7) Lines 26-29, page 9: The authors may also wish to note that in EPA's 2015 review of the ozone standard, they considered potential secondary W126 ozone standard levels between 7 and 17 ppm-hrs and the Clean Air Science Advisory Committee recommended a W126 standard level between 7 and 15 ppm-hrs (See EPA, 2015). These levels are consistent with the levels discussed here from other literature sources.**

*We have added text to this point.*

Page 10, Lines 14–16: "The EPA has considered a potential secondary W126 ozone standard between 7 and 17 ppm h (Environmental Protection Agency, 2015a); likewise, the Clean Air Science Advisory Committee recommended a W126 standard level between 7 and 15 ppm h (Environmental Protection Agency, 2015a)."

**8) Lines 24-25, page 11: There are more recent EPA regulations on heavy-duty onroad and nonraod emissions that could be cited here. See: https://www.epa.gov/emission-standards-reference-guide/epa-emission-standards-regulations**

*We have updated references at both the federal and state levels.*

Page 12, Lines 15–19: "Local $NO_x$ emissions should continue to decline into the future, as there are significant controls currently ongoing or in the implementation phase, including more stringent national rules on heavy-duty diesel engines (Environmental Protection Agency, 2000; 2010), combined with California Air Resources Board (CARB) diesel engine retrofit-replacement requirements (California Air Resources Board, 2008; 2014), and more stringent CARB standards for gasoline-powered vehicles (California Air Resources Board, 2012)."

**9) Lines 3-23, page 12: Authors may want to note that less substantial trends in the spring may also be due to a larger fraction of ozone coming from background ozone sources in the spring than in the summer. The authors discuss background ozone in the next paragraph but never explicitly state this.**

*We have stated this more explicitly.*

Page 13, Lines 13–15: "Deeper cuts in emissions appear to be required in the springtime in SNP, as decreases in anthropogenic emissions have a smaller effect, both relatively and in the absolute, on the total $O_3$ abundance than during $O_3$ season, in part because background $O_3$ makes the greatest contribution to daily $O_3$ in the springtime SNP (Figure 4)."

**10) Line 5, page 13: Also note, that satellite observations as well as Chinese emissions estimates indicate that Chinese NOx emissions have been decreasing since 2011 so the influence from Asia may have become less important since 2011.**

*We have included this information.*

Page 13, Lines 29–33: "East Asian $NO_x$ emissions have risen over our study window (e.g., Miyazaki et al., 2017), potentially causing an increase in the influence of trans-Pacific transport on $O_3$ concentrations at SEQ2 and reducing the efficacy of local $NO_x$ control in springtime. However, $NO_x$ emission and concentration declines have been observed over China since 2011 (Liu et al., 2016), potentially diminishing the influence of Asian transport events in SNP."

**11) Figure 6: in caption or figure headings clarify that MD8A is the seasonal average.**

*We have clarified this.*

Page 27, Lines 7–10: "**Figure 6.** $O_3$ trends in Visalia (orange diamonds), SEQ1 (cyan filled circles), and SEQ2 (dark blue open circles) computed using MD8A (a–b), SUM0 (c–d), W126 (e–f), and morning $O_x$ (g–h) metrics during $O_3$ season (top row) and springtime (bottom row). Both MD8A and morning $O_x$ are

computed as seasonal averages. Error bars in panels a–b and g–h are standard errors of the mean. Error bars in panels c and e are standards errors of the mean of the three $O_3$ season 3-month summations."

[revised manuscript text omitted]